

**Pre-monsoon air quality over Lumbini, a world heritage site**
**along the Himalayan foothills**
Dipesh Rupakheti[1,2*], Bhupesh Adhikary[3], Puppala S. Praveen[3], Maheswar Rupakheti[4,5],
Shichang Kang[6,7*], Khadak S. Mahata[4], Manish Naja[8], Qianggong Zhang[1,7], Arnico K. Panday[3],
Mark G. Lawrence[4]
[1]Key Laboratory of Tibetan Environment Changes and Land Surface Processes, Institute of
Tibetan Plateau Research, Chinese Academy of Sciences, Beijing 100101, China
[2]University of Chinese Academy of Sciences, Beijing 100049, China
[3]International Centre for Integrated Mountain Development (ICIMOD), Kathmandu, Nepal
[4]Institute for Advanced Sustainability Studies (IASS), Potsdam 14467, Germany
[5]Himalayan Sustainability Institute (HIMSI), Kathmandu, Nepal
[6]State Key Laboratory of Cryospheric Science, Cold and Arid Regions Environmental and
Engineering Research Institute (CAREERI), Lanzhou 730000, China
[7]Center for Excellence in Tibetan Plateau Earth Sciences, Chinese Academy of Sciences, Beijing
100085, China
[8]Aryabhatta Research Institute of Observational Sciences (ARIES), Nainital, India
*Correspondence to*:
D. Rupakheti (dipesh.rupakheti@itpcas.ac.cn), S.C. Kang (shichang.kang@lzb.ac.cn )



**Abstract**
Lumbini, in southern Nepal, is a UNESCO world heritage site of universal value as the
birthplace of Buddha. Poor air quality in Lumbini and surrounding regions is a great concern for
public health as well as for preservation, protection and promotion of Buddhist heritage and
culture. We present here results from measurements of ambient concentrations of key air
pollutants (PM, BC, CO, $O_3$) in Lumbini, first of its kind for Lumbini, conducted during an
intensive measurement period of three months (April-June 2013) in the pre-monsoon season. The
measurements were carried out as a part of the international air pollution measurement
campaign; SusKat-ABC (Sustainable Atmosphere for the Kathmandu Valley - Atmospheric
Brown Clouds). The ranges of hourly average concentrations were: $PM_{10}$: 10.5 - 604.0 µg m$^{-3}$,
$PM_{2.5}$: 6.1 - 272.2 µg m$^{-3}$; BC: 0.3 - 30.0 µg m$^{-3}$; CO: 125.0 - 1430.0 ppbv; and $O_3$: 1.0 - 118.1
ppbv. These levels are comparable to other very heavily polluted sites throughout South Asia.
The 24-h average $PM_{2.5}$ and $PM_{10}$ concentrations exceeded the WHO guideline very frequently
(94% and 85% of the sampled period, respectively), which implies significant health risks for the
residents and visitors in the region. These air pollutants exhibited clear diurnal cycles with high
values in the morning and evening. During the study period, the worst air pollution episodes
were mainly due to agro-residue burning and regional forest fires combined with meteorological
conditions conducive of pollution transport to Lumbini. Fossil fuel combustion also contributed
significantly, accounting for more than half of the ambient BC concentration according to
aerosol spectral light absorption coefficients obtained in Lumbini. WRF-STEM, a regional
chemical transport model, was used to simulate the meteorology and the concentrations of
pollutants. The model was able to reproduce the variation in the pollutant concentrations well;
however, estimated values were 1.5 to 5 times lower than the observed concentrations for CO
and $PM_{10}$ respectively. Regionally tagged CO tracers showed the majority of CO came from the
upwind region of Ganges valley. The model was also used to examine the chemical composition
of the aerosol mixture, indicating that organic carbon was the main constituent of fine mode
$PM_{2.5}$, followed by mineral dust. Given the high pollution level, there is a clear and urgent need
for setting up a network of long-term air quality monitoring stations in the greater Lumbini
region.



## 1. **Introduction**

The Indo-Gangetic plain (IGP) stretches over 2000 km encompassing a vast area of land in northern South Asia: the eastern parts of Pakistan, most of northern and eastern India, southern part of Nepal, and almost all of Bangladesh. The Himalayan mountains and their foothills stretch along the northern edge of IGP. The IGP region is among the most fertile and most intensely farmed region of the world. It is a heavily populated region with about 900 million residents or 12% of the world's population. Four megacities - Lahore, Delhi, Kolkata, and Dhaka are located in the IGP region, with dozens more cities with populations exceeding one million. The region has witnessed impressive economic growth in recent decades but unfortunately it has also become one of the most polluted, and an air pollution 'hot spot' of local, regional and global concern (Ramanathan et al., 2007). Main factors contributing to air pollution in the IGP and surrounding regions include emissions from vehicles, thermal power plants, industries, biomass and fossil fuel used in cooking and heating activities, agricultural activities, crop residue burning and forest fires. Air pollution gets transported long distances away from emission sources and across national borders. As a result, the IGP and adjacent regions get shrouded with a dramatic annual buildup of regional scale plumes of air pollutants, known as Atmospheric Brown Clouds (ABC), during the long and dry winter and pre-monsoon seasons each year (Ramanathan and Carmichael, 2008). Figure 1 shows the mean aerosol optical depth (AOD) acquired with the MODIS instrument onboard TERRA satellite over South Asia for a period of December 2012- June 2013. Very high aerosol loading along the entire stretch of IGP reflects severity of air pollution over large area in the region.

Poor air quality continues to pose significant threat to human health in the region. In a new study of global burden of disease released recently, Forouzanfar et al. (2015) estimated that in 2013 around 1.7 million people died prematurely in Pakistan, India, Nepal, and Bangladesh as a result of air pollution exposure, nearly 30% of global total premature deaths due to air pollution. Air pollution also affects precipitation (e.g. South Asian monsoon), agricultural productivity, ecosystems, tourism, climate, and broadly socio-economic and national development goals of the countries in the region (Burney and Ramanathan, 2014; Shindell, 2011; Ramanathan and Carmichael, 2008). It has also been linked to intensification of cold wave and winter fog in the IGP region over recent decades (Lawrence and Lelieveld, 2010 and references therein; Safai et





al., 2009; Ganguly et al., 2006). Besides high levels of aerosol loading as shown in Fig. 1, Indo-
Gangetic plains also have very high levels of ground level ozone or tropospheric ozone ($O_3$)
(e.g., Ramanathan and Carmichael (2008)). It is a toxic pollutant to plant and human health, and
a major greenhouse gas (IPCC, 2013; Shindell, 2011; Mohnen et al., 1993). South Asia, in
particular IGP region, has been projected to be most ozone polluted region in world by 2030
(Stevenson et al., 2006). Majority of crop loss in different parts of the world results from effects
of ozone on crop health and productivity (Shindell, 2011). For example, Burney and Ramanathan
(2014) reported a significant loss in wheat and rice yields in India from 1980 to 2010 due to
direct effects of black carbon (BC) and ozone ($O_3$). BC and $O_3$ are two key short-lived climate
pollutants (SLCP). Because of the IGP's close proximity to the Himalaya-Tibetan plateau region,
this once relatively clean region, is now subjected to increasing air pollution transported from
regions such as the IGP, which can exert additional risks to human health and sensitive
ecosystems in the mountain region (e.g., (Lüthi et al., 2015; Marinoni et al., 2013; Duchi et al.,
2011). Studies have shown elevated BC loading over the Himalaya-Tibetan region results in
additional atmospheric warming which combined with  BC deposition on snow and ice leads to
accelerated melting of the snow and glaciers (Shindell, 2011; Xu et al., 2009; Ramanathan and
Carmichael, 2008). Air pollution transport pathways to Himalayas are still not yet fully
understood.
Air pollution can also damage the built environment and cultural and archeological heritages
(Brimblecombe, 2003). Monuments and buildings made with stones are vulnerable to air
pollution damage (Brimblecombe, 2003; Gauri and Holdren, 1981). Sulfur dioxide, which forms
sulfuric acid upon reaction with water is the most harmful substance for the monuments as it can
corrode and damage them (Baedecker et al., 1992; Gauri and Holdren, 1981). Indo-Gangetic
plains are rich in archeological, cultural and historical sites and monuments and many of them
are inscribed as UNESCO World Heritage Site. For example, among many other such sites in
IGP are the Archaeological Ruins at Moenjodaro (Pakistan), Taj Mahal in Agra and Mahabodhi
Temple Complex in Bodh Gaya (India), Lumbini (Nepal), and ruins of the Buddhist Vihara at
Paharpur (Bangladesh) (World Heritage List; UNESCO, website: http://whc.unesco.org/en/list).
The Taj Mahal is one of the seven wonders of the modern world and India's greatest landmark.
Starting in 1970s, there have been observations of brownish/yellowish tone on its shiny



white marble façade, and the primary suspect of discoloration was heavy air pollution from
industries and traffic that grew around the monument site in Agra over the past decades. At the
end of the last century, the government of India realized the growing problem and started a
program to save the monument. It introduced measures to cut back pollution, as well as set up
stations around the monument to monitor air quality around the clock. A recent study has
reported that deposition of light absorbing aerosol particles (black carbon, brown carbon) and
dust is responsible for its discoloration (Bergin et al., 2015).
Lumbini, located near the northern edge of the central Ingo-Gangetic plain, is famous as the
birthplace of the Lord Buddha. Lumbini is a UNESCO world heritage site of outstanding
universal value to humanity, inscribed in the UNESCO list since 1997. The site, with valuable
archaeological remains of the Buddhist *Viharas* (monasteries) and *Stupas* (memorial shrines), as
well as modern temples and monasteries, is a center of attraction and visited by hundreds of
thousands of pilgrims, scientists, scholars, yogis, and tourists every year. Over recent years, there
is increasing concern about poor air quality in Lumbini and the surrounding region. There is no
surface monitoring of air quality in Lumbini.
As a first attempt to understand air quality in Lumbini, we carried out continuous measurements
of ambient concentrations of key air pollutants (particulate matter, black carbon, carbon
monoxide, ozone) and other auxiliary measurements (Aerosol optical depth – not discussed on
the present study, meteorological parameters) during an intensive measurement period of three
months (April-June) in the year 2013. These are the first reported measurements for Lumbini. A
regional chemical transport model called Sulfur Transport and dEposition Model (STEM) was
used to simulate the variations of meteorological parameters and air pollutants during the
observation period. Regionally tagged CO tracers were used to identify emission source regions
impacting pollutant concentration observed at Lumbini. Satellite data has also been used to
understand the high pollution events during the monitoring period. These measurements were
carried out as a part of the SusKat-ABC international air pollution measurement campaign (*M.*
*Rupakheti, manuscript in preparation for ACPD*) jointly led by the International Centre for
Integrated Mountain Development (ICIMOD), Kathmandu, Nepal and Institute for Advanced
Sustainability Studies (IASS), Potsdam, Germany.





## 2. Experimental set up

### 2.1 Sampling site

The Lumbini measurement site (27º29.387′ N, 83º16.745′ E, elevation: ~100 m above sea level) is located at the premise of the Lumbini International Research Institute (LIRI), a Buddhist library in Lumbini. Lumbini lies in the Nepal's southern lowland plain or *Terai* region, termed as "bread basket of Nepal" due to the availability of very fertile land suitable for crop production, which forms the northern edge of the Indo-Gangetic Plains (IGP). 25 km north of Lumbini the foothills begin, while the main peaks of the Himalayas are 140 km to the north. The remaining three sides are surrounded by flat plain land of Nepal and India. The site is only about 8 km from the Nepal-India boarder in the south. A three storied 10 m tall water tower was used as the platform for the automatic weather station (AWS) whereas remaining instruments were placed inside a room near the base of the tower. Figure 2 shows the location of Lumbini, the Kenzo Tange Master Plan area of the Lumbini development project, and the sampling tower. An uninterrupted power back up was set up in order to assure the regular power supply even during hours with scheduled power cuts during the monitoring period. The nearby premises of the monitoring site consist of the LIRI main office and staff quarters. Further away is a museum, a local bus park for the visitors to Lumbini, the office of the Lumbini Development Trust, monasteries, and thinly forested area with grassland within the master plan area. Outside of the master plan area lie vast area of agricultural fields, village pockets, and several brick kilns and cement industries. A local road (black topped), that cuts through the master plan area, lies about 200 m north of the sampling site and experiences intermittent passing of vehicles. According to the Ministry of Culture, Tourism and Civil Aviation of Nepal over 130 thousand tourists (excluding Nepalese and Indian citizens) visited the Lumbini area in 2014 (http://tourism.gov.np/en).

### 2.2 Monitoring Instruments

The summary of instruments deployed in Lumbini is presented in Table 1. They monitored ambient concentrations of various air pollutants and local meteorological parameters continuously during the sampling period of about two and half months. All data were collected in Nepal Standard Time (NST) which is GMT +05:45 hour. $PM_{10}$, $PM_{2.5}$ and $PM_1$ mass





concentrations were monitored continuously with GRIMM EDM164 (GRIMM Aerosol Technik,
Germany), reporting data every 5 min. The instrument uses the light scattering at 655 nm to
derive mass concentrations. More description on the technical aspects of the instrument can be
found on the manufacturer's website (http://wiki.grimm-aerosol.de/index.php?title=ENVIRO-
EDM164). The EDM164 used in this study was a newly purchased instrument which was
calibrated at the factory of the GRIMM Aerosol Technik in Germany before it was deployed at
Lumbini. Similarly, aerosol light absorptions at 7 wavelengths (370, 470, 520, 590, 660, 880,
950 nm) were measured continuously with an Aethalometer (Model AE-42, Magee Scientific,
USA), averaging and reporting data every 5 min. AE-42 was operated at a flow rate of 5 l min$^{-1}$.
As described by the manufacturer, ambient BC concentration is derived from light absorption at
880 nm using a specific mass absorption cross section. To obtain BC concentration in Lumbini,
we used a specific mass absorption cross-section value of 8 m$^2$ g$^{-1}$ for the 880 nm channel.
Similar value has been previously used for BC measurement in the Indo-Gangetic plain (Praveen
et al., 2012). Optical measurement by filter-based absorption photometers, such as the
Aethalometer, suffer from measurement artifact known as filter loading effect which must be
taken into account and corrected for while deriving ambient BC concentrations. We used
correction method suggested by Schmid et al. (2006) which was also used by Praveen et al.
(2012) for BC measurements at a rural site in the Indo-Gangetic plain. Surface ozone ($O_3$)
concentration was measured continuously with an ozone analyzer (Model 49$i$, Thermo Scientific,
USA), reporting data every minute. It utilizes UV (254 nm wavelength) photometric technology
to measure ozone concentration in ambient air. CO analyzer (Model 48$i$, Thermo Scientific,
USA) was used to monitor ambient CO concentrations, recording data every minute. The CO
analyzer is based on the principle that CO absorbs infrared radiation at the wavelength of 4.6
microns. The ambient air was drawn through 6-micron pore size SAVILLEX 47 mm filter at the
inlet in order to remove the dust particles before sending air into the CO and $O_3$ analyzers using a
Teflon tube. The filters were replaced every 7-10 days depending on particle loading, based on
manual inspection. Both CO and $O_3$ analyzers were new instruments, freshly calibrated at the
factory before deploying them in Lumbini. The CO instrument was set to auto-zero at a regular
interval of 6 hours. Local meteorological parameters (temperature, relative humidity, wind
speed, wind direction, precipitation, and global solar radiation) were monitored with an





automatic weather station (AWS) (Campbell Scientific, Loughborough, UK), recording data
every minute.

## 2.3 Regional chemical transport model

Aerosol and trace gas distributions were simulated using a regional chemical transport model.
Sulfur Transport and dEposition Model (STEM), a 3D eulerian model, that has been used
extensively in the past to characterize air pollutants in South Asian region was used to interpret
observations at Lumbini (Kulkarni et al., 2015; Adhikary et al., 2007). The Weather Research
and Forecasting (WRF) model (Skamarock et al., 2008) version 3.5.1 was used to generate the
required meteorological variables necessary for simulating pollutant transport in STEM. The
model domain was centered at 24.94º N latitude and 82.55º E longitude covering a region from
3.390º N to 43.308º N latitudes and 34.880º E to 130.223º E longitudes. The model has $425\times200$
horizontal grid cells with grid resolution of $25\times25$ km and 41 vertical layers with top of the
model set at 50 mbar. The WRF model was run from November 1, 2012 to June 30, 2013.
However, for this study, modeled data only from April to June 2013 have been used. The WRF
model was initialized with FNL data available from NCAR/UCAR site
(http://rda.ucar.edu/datasets/ds083.2/).
The tracer version of the STEM model provides mass concentration of sulfate, BC (hydrophilic
and hydrophobic), Organic carbon (OC), sea salt (fine and coarse mode), dust (fine $PM_{2.5}$ and
$PM_{10}$), CO (biomass and anthropogenic) and region tagged CO tracers. STEM model domain
size, resolution and projection are those of the WRF model. Details about tracer version of the
STEM model is outlined elsewhere (Kulkarni et al., 2015; Adhikary et al., 2007).
Anthropogenic emission of various pollutants ($CH_4$, CO, $SO_2$, $NO_x$, NMVOC, $NH_3$, $PM_{10}$,
$PM_{2.5}$, BC and OC) used in this analysis were taken from the EDGAR-HTAP_v2
(http://edgar.jrc.ec.europa.eu/htap_v2/index.php?SECURE=123). Emission inventory were
developed for the year 2010 gridded at the spatial resolution of 0.1º×0.1º. Open biomass burning
emissions on a daily basis during the simulated period were taken from data obtained from the
FINN model (Wiedinmyer et al., 2011). As with the WRF model, the STEM model was run from
November 2, 2012 to June 30, 2013 however, data presented here are only during the intensive
field campaign period.





3   **Results and discussions**
**3.1 Meteorology**
**3.1.1   Time series of local meteorological parameters**
Hourly average time series of various meteorological parameters viz. precipitation in mm hr$^{-1}$
(Prec), relative humidity in % (RH), temperature in ºC (T), wind direction in degree (WD) and
wind speed in m s$^{-1}$ (WS) during the monitoring period are shown in Figure 3. Meteorological
parameters were obtained with the sensors at the height of ~12 m from the ground. Moreover,
meteorology results from simulations using a 3D model have been used to compare with the
observations, and to fill the data gaps. Precipitation data was derived from TRMM satellite
(TRMM_3B42_007 at a horizontal resolution of 0.25º) from the Giovanni platform
(http://giovanni.gsfc.nasa.gov/giovanni/) as the rain guage malfuntioned during the sampling
period.
Average observed wind speed during the study period was 2.4 m s$^{-1}$, with hourly values ranging
between 0.03 - 7.4 m s$^{-1}$ whereas from the WRF model average wind speed was found to be 3.2
m s$^{-1}$ (range: 0.06 - 11.1 m s$^{-1}$). Comparison of the model output data with observation shows
that the model adequately captures wind speed to study pollutant transport. Diurnal variation of
observed hourly average wind speed suggested that wind speeds were lower during nights and
mornings while higher wind speed prevailed during day time, with average winds > 3 m s$^{-1}$ up to
~ 3.3 m s$^{-1}$ between 09:00-13:00 local time (Supplementary materials, Figure S1, upper panel).
High speed strong winds (> 4 m s$^{-1}$) were from the NW direction during the month of April
which later switched to almost opposite direction, i.e., SE direction from the month of May
onwards. Figure 4 shows the monthly wind rose plot (using WRPLOT view from the Lakes
Environmental, http://weblakes.com/). Average observed temperature for the sampling period
until the sensor stopped working (on 8$^{th}$ May, 2013, i.e., for 38 days) was 28.1ºC, with a
minimum value recorded to be 16.5ºC whereas the maximum was 40ºC. Average T from the
model, during same period, was 31ºC with values ranging between 19 - 40ºC.  The model
captures the synoptic variability of temperature and is mostly within the range of daily values.
However, the model has a high bias and does not capture daily minimum temperature values. For
the same period (until the sensor stopped working), the average (observed) RH was ~ 50%



(ranging from 10.5 to 97.5%) whereas the model showed the average RH to be ~ 23% (same
period as observation) with values ranging between 6 to 78%. RH values are highly
underestimated by the model however; the synoptic scale variability is captured by the model.
Discrepancy on model results might have occurred due to various factors inherently uncertain in
a weather model. However, we believe that modeled data is vital for understanding pollutant
transport in an area where observation data are non-existent or are incomplete.
**3.1.2   Synoptic scale winds during pre-monsoon**
The monthly mean synoptic wind for the month of April, May and June is presented in Figure 5.
NCEP/NCAR reanalysis monthly data of winds at 1000 mbar were used to study the wind
pattern. The red dot in the figure indicates the location of Lumbini. NCEP/NCAR data showed
the dominance of calm winds over the measurement site. Similar type of wind directions were
observed over Kanpur, India, also in the IGP, during the pre-monsoon season (Srivastava et al.,

270   2011).


**3.2 Time series of air pollutants**
**3.2.1   General overview**
Figure 6 shows hourly averaged time series of both observed and modeled $PM_{10}$, $PM_{2.5}$, BC, CO
and $O_3$ observed at Lumbini during the study period. In this section, results have been discussed
based upon the observation datasets only. Section 3.2.3 will discuss model comparison and
interpretation.
*Both PM fractions:* $PM_{10}$ and $PM_{2.5}$ showed similar temporal behavior. Observed hourly average
$PM_{10}$ concentrations ranged between 10.5-603.9 µg m$^{-3}$ with an average of 128.9±91.9 µg m$^{-3}$
whereas $PM_{2.5}$ concentrations ranged between 6.1 and 272.2 µg m$^{-3}$ with an average of 53.1±35.1
µg m$^{-3}$ during the sampling period. In addition to this, average $PM_1$ concentration was 35.8±25.6
µg m$^{-3}$ with the concentrations ranging between 3.6 to 197.6 µg m$^{-3}$. $PM_1$ concentration has not
been discussed in this study. The observed 24 hour average particulate matter concentrations
($PM_{2.5}$ and $PM_{10}$) were found frequently higher than the WHO prescribed guidelines for $PM_{2.5}$
(25 µg m$^{-3}$) and $PM_{10}$ (50 µg m$^{-3}$), (WHO, 2006) –   $PM_{2.5}$: 94% and $PM_{10}$: 85% of the



measurement period of 53 days. Similarly, BC concentrations during the measurement period
ranged between 0.3-29.9 μg m$^{-3}$ with a mean (±SD) value of 4.9 (±3.8) μg m$^{-3}$. The lowest
concentration was observed during a rainy day (21-22 April) whereas the highest concentration
was observed during a period of forest fire (detailed in Section 3.4). BC concentrations in
Lumbini during pre-monsoon months are lower compared to BC concentrations observed in the
Kathmandu Valley because of high number of vehicles plying on the street, brick kilns and other
industries in Kathmandu valley (Putero et al., 2015; Sharma et al., 2012).
CO concentrations ranged between 124.9-1429.7 ppbv with an average value of 344.1±160.3
ppbv. CO concentration observed in Lumbini is lower than that of Mohali, Western India where
the average concentration was 566.7 ppbv during pre-monsoon season due to intense biomass
and agro-residue burning over the region (Sinha et al., 2014). Temporal variation of CO
concentrations is similar to that of BC as both of these species are emitted during incomplete
combustion of fuel. BC to CO ratio in Lumbini was found to be different from that observed at
other urban and rural sites and those affected by forest fire/biomass burning. However, a sub-
urban site, Pantnagar, in IGP also observed similar BC to CO ratio (Joshi et al., 2016) as
observed in Lumbini. There was a very strong correlation (r > 0.9) between BC and CO
(Supplementary material, Figure S2), indicating likely common sources of emission for both
pollutants. The hourly averaged observed ozone concentration ranged between 1.0 and 118.1
ppbv with a mean value of 46.6±20.3 ppbv during the sampling period. The 8-hr maximum O$_3$
concentration exceeded WHO guidelines of 100 μg m$^{-3}$ (WHO, 2006) during 88% of the
measurement period. Our results clearly indicate that the current pollution levels in Lumbini is of
great concern to health of the people living in the region as well as over a million visitors who
visit Lumbini, as well as ecosystems, particularly agro-ecosystem, especially in warm and sunny
pre-monsoon months.
**3.2.2   Comparison with other south Asian sites**
Past studies near this site have been focused on the cities like Kathmandu (Putero et al., 2015;
Sharma et al., 2012; Ram et al., 2010; Panday and Prinn, 2009) and Kanpur (Ram et al., 2010)
and agro-residue burning dominated regions of IGP (Rastogi et al., 2016; Sinha et al., 2014;
Sarkar et al., 2013) or a remote mountain location in India (Naja et al., 2014). In order to put our



results in perspective, pollution levels observed in Lumbini have been compared with the
observations from other sites in the region and are presented in Table 2. Very high aerosol
loading is observed in South Asia during pre-monsoon, mostly over the IGP region
(Supplementary materials, Figure S3). $PM_{2.5}$ concentration in Lumbini have been found to be
lower than the megacity like Delhi (Bisht et al., 2015) and north-western IGP regions (Sinha et
al., 2014) due to higher level of emissions (from traffic and biomass burning respectively) over
those regions. BC concentrations observed in Lumbini during pre-monsoon season was lower
than the urban Asian cities like Kathmandu (Putero et al., 2015) and Delhi (Bisht et al., 2015),
slightly higher than in Kanpur but high compared to the remote locations in the region. BC
observed at Lumbini was higher by a factor of ~6 and ~4.5 compared to that at Mt. Abu, India
(Das and Jayaraman, 2011) and near the base of Mt. Everest, Nepal (Marinoni et al., 2013)
respectively. Regarding CO, concentration in Lumbini was ~ 1.5-5 times lower than other urban
locations in India (Gaur et al., 2014; Sinha et al., 2014). However, Lumbini CO concentrations
are ~2.3-2.6 times higher than nearby remote location such as Mt. Abu (Naja et al., 2003).
Average $O_3$ concentrations, over sampling period, in Lumbini were found to be higher than the
cities like Kathmandu (Putero et al., 2015). However, ozone concentrations higher than that
observed at Lumbini were reported at nearby city of Kanpur during pre-monsoon season (Gaur et
al., 2014). Interestingly ozone concentrations higher than that at Lumbini were observed in the
Mt. Everest region. Uplift of the polluted air masses (Marinoni et al., 2013), stratospheric
intrusion (Cristofanelli et al., 2010) and even the regional or long-range transport of the air
pollutants (Bonasoni et al., 2010) might have contributed for the higher ozone concentration over
the Everest region, resulting in higher $O_3$ concentration compared to Lumbini.
**3.2.3  Observation-model inter-comparison**
Chemical transport models provide insight to observed phenomena; however, interpretation has
to take into account model performance before arriving at any conclusion. This section describes
pollution concentrations simulated by the WRF-STEM model. A comparison of model calculated
average concentration along with the minimum and maximum concentrations of various
pollutants (with observation) is shown in Table 3. The model based concentrations used here are
instantaneous values for every third hour of the day. Regarding $PM_{2.5}$ and $PM_{10}$, the model
simulated average concentration was 17.3±6.7 (1.9-48.3) µg m$^{-3}$ and 25.4±12.9 (2.1-68.8) µg m$^{-}$





$^3$, respectively. The model estimated values were lower by the factor of 3 and 5 respectively than
the observed concentrations. Similarly, average CO concentration was 255.7±83.5 ppbv, ranging
between 72.2-613.1 ppbv, with average model CO ~1.35 times lower than observed. BC
concentrations ranged between 0.4-3.7 µg m$^{-3}$ with a mean value of 1.8±0.7 µg m$^{-3}$ for a period
of $1^{st}$ April-$15^{th}$ June 2013. The average model BC concentration was ~2.7 times lower than the
observed BC. Previous study using the STEM model over Kathmandu valley showed the model
was able to capture annual BC mean value but completely missed the concentrations during pre-
monsoon and post monsoon period (Adhikary et al., 2007). Similar behavior is seen this time for
CO where the model misses the peak values but reasonably captures CO concentration after mid-
May. Even though the model calculated values are lower in the present study, the model captures
the synoptic variability fairly well for all the pollutants compared. STEM model performance can
be significantly improved via better constraining anthropogenic emissions inventory, emissions
of open biomass burning (natural and anthropogenic) and improvements in meteorological
output from WRF amongst many other uncertainties inherent in regional chemical transport
model. This activity is beyond the scope of this current paper although the improvements are
underway for all these sectors.

### Diurnal variations of air pollutants and boundary layer height

In the emission source region, diurnal variations of primary pollutants provide information about
the time dependent emission activities (Kumar et al., 2016). Figure 7 shows the diurnal variation
of hourly averaged concentrations of various pollutants measured during the sampling period.
Primary pollutants like PM$_{10}$, CO and BC all showed typical characteristics of an urban
environment, i.e., diurnal variation with a morning and an evening peak. However, Lumbini data
shows higher concentrations in the evenings compared to morning hours. Elevated
concentrations can be linked to morning and evening cooking hours for BC and CO where
emission inventory show that residential sector has significant contribution. However,
explanation for elevated evening concentration compared to morning needs further investigation.
Increase in the depth of boundary layer, reduction in the traffic density on the roads, absence of
open biomass burning during mid-day and increasing wind speed often contribute to the
dispersion of pollutants resulting in lower concentration during afternoon. Diurnal variation of
wind direction (Supplementary information, Figure S1, lower panel) shows the dominance of





wind coming from south (mainly during the month of May and till mid-June). Morning and
evening period experienced the winds coming from the southeast direction while the winds were
predominantly from southwest direction during late afternoon. Increase in CO concentrations in
the evening hours might be due to transport of higher levels of CO emissions from source
regions upwind of Lumbini which along with the local emissions gets trapped under lower
Planetary Boundary Layer (PBL) heights in evening and night time.  Ozone concentration was
lowest in the morning before the sunrise and highest in late afternoon around 15:00 PM after
which concentrations started declining, exhibiting a typical characteristic of a polluted urban site.
Photo-dissociation of accumulated $NO_x$ reservoirs (like HONO) provides sufficient NO
concentration leading to the titration of $O_3$ resulting in minimum $O_3$ just before sunrise (Kumar
et al., 2016). The PBL height (in meters (m)) was obtained from the model as observations were
not available. Figure 8 shows the diurnal variation of the model derived PBL height. The study
period average PBL height over Lumbini was ~ 910 m (ranging between 24.28 and 3807 m
observed at 06:00 and 15:00 h respectively). As the pre-monsoon month advances, PBL height
also increased. The monthly average PBL height was 799 m, 956 m and 1014 m respectively
during the month of April, May and (1st-15th) June. Over the IGP region, PBL height is deeper
during the pre-monsoon compared to monsoon (Patil et al., 2014), post-monsoon (Hegde et al.,
2009) and winter (Badarinath et al., 2009) seasons. The fluctuations of PBL height correspond
well with the diurnal variation of the pollutants like BC, CO and PM with the period of lower
boundary height experiencing higher pollution concentration.

### 3.3 Influence of forest fires on Lumbini air quality

### 3.3.1  Identification of forest fire influence over large scale using in-situ observations satellite and model data

Forest fires and biomass burning (mostly agro-residue burning in large scale) are common over
the South Asia and the IGP region during pre-monsoon season. North Indo-Gangetic region is
characterized by fires even during the monsoon and post-monsoon season (Kumar et al., 2016;
Putero et al., 2014). These  activities influence air quality not only over nearby regions but also
get transported towards high elevation pristine environments like Everest (Putero et al., 2014)
and Tibet (Cong et al., 2015a; 2015b). So, one of the main objectives of this study was to





identify the influence of open burning on Lumbini air quality. Average wind speed during the
whole measurement period was 2.4 m s$^{-1}$. Based on this data, open fire counts within the grid
size of 200×200 km centering over Lumbini was used for this analysis assuming that the
emissions will take a maximum period of one day to reach our monitoring site. Forest fire counts
were obtained from MODIS satellite data product called Global Monthly Fire Location Products
(MCD14ML). More on this has already been described by Putero et al. (2014). Figure 9 shows
the daily average in-situ CO, BC, aerosol absorption Ångstrom exponent (AAE) which is derived
from Aethalometer data and daily open fire count within the specified grid. The green box in the
figure is used to show two outstanding events with the elevated BC and CO concentrations
observed during the monitoring period. The first peak was observed during 7-9 April and second
peak during 3-4 May, 2013. Two pollutants having biomass burning as the potential primary
source: BC and CO were taken in consideration. AAE values higher during these two events (~
1.6) are also an indication of presence of BC of biomass burning origin. Ground based TSP
sampling also showed higher concentration of biomass burning tracer (potassium or K$^+$) in
Lumbini during the pre-monsoon season comparing to other seasons of the year (*L. Tripathee,*
*personal communication*). But, to our expectation, we could not observe any significant
influence of forest fire within the specified grid (or the influence of local forest fire on the air
quality over Lumbini was not observed). Therefore, a wider area, covering South and Southeast
Asian regions, was selected for the forest fire count. Figure 10 (A-B) shows the active fire
hotspots from MODIS, over the region, during the peak events which shows the first peak
occurred due to the forest fire over the eastern India region whereas the second peak was
influenced by the forest fire over western IGP region. Moreover, in order to strengthen our
hypothesis, we have utilized satellite data products for various gaseous pollutants like CO and
NO$_2$ (Atmospheric Infrared Sounder (AIRS) for CO and Ozone Monitoring Instrument (OMI) for
NO$_2$ both obtained from Giovanni platform). Figure 10 (C-H) shows the daytime total column
CO before, during and after occurrence of two events (peaks) as stated earlier. Atmospheric
Infrared Sounder (AIRS) satellite with daily temporal resolution and 1°×1° spatial resolution
have been utilized to understand the CO concentration over the area. CO concentration over
Lumbini during both of the peaks confirmed the role of open fires on either sides of the IGP
region for elevated concentration of CO over Lumbini. To further strengthen our finding, the aid
of wind rose plot of local wind speed and direction was taken. Figure 10 (I-J) represent the wind



rose plot only for these two events respectively. Wind rose plots also confirm the wind blowing
from those two forest fire regions affected the air quality in Lumbini region. Figure10 (K) shows
model biomass CO peak coincident with observed CO. Although the magnitudes are different,
the timing of the peaks is well captured by the model. However, satellite based open fire
detection also has limitation as it does not capture numerous small fires that are prevalent over
south Asia which usually burn out before the next satellite overpass. More research is needed to
assess the influence of these small fires on regional air quality.
In a separate analysis (not shown here), elevated $O_3$ concentration during these two events were
also observed. Average $O_3$ concentration before, during and after the events were found to be
46.2±20.3 ppbv, 53.5±31.1 ppbv and 50.3±20.9 ppbv respectively (Event-I) whereas it was
found to be 54.8±23.8 ppbv, 56.7±35 ppbv and 55.6±13.4 ppbv respectively (Event-II).
Increased ozone concentrations during the high peak events have been analyzed using the
satellite $NO_2$ concentration over the region considering the role of $NO_2$ as precursor for ozone
formation. Daily total column $NO_2$ were obtained from OMI satellite (data available at the
Giovanni platform; http://giovanni.gsfc.nasa.gov/giovanni/) at the spatial resolution of
0.25º×0.25º. Figure 11 shows the $NO_2$ column value before, during and after both events. Even
for the $NO_2$, maximum concentrations were observed during these two special events.
**3.3.2 Identifying regional contribution**
An attempt has been undertaken to identify the source region contribution, utilizing the WRF-
STEM model results, for the CO concentrations observed at Lumbini. A recent study (Kulkarni
et al., 2015) has explored the source region contribution of various pollutants over the Central
Asia using the same model. Figure 12 (A) shows the average contribution from different regions
on CO concentration over Lumbini during the whole measurement period. Major share of CO
was from the Ganges valley (46%) followed by Nepal region (25%) and rest of Indian region
(~17.5%). Contribution from other South Asian countries like Bangladesh and Pakistan were ~
11% whereas China contributed for ~1% of the CO concentration in Lumbini.
Figure 12 (B) is the time series of percentage contribution to total CO concentration during
whole measurement period showing different air mass arriving at a 3 hourly intervals. During the
whole measurement period, majority of the CO reaching Lumbini were from the Ganges valley





region with the contribution sometimes reaching up to ~80%. Other India (central, south, east
and north) regions also contributed significantly. Bangladesh's contribution in CO loading was
seen only after mid-April lasting for only about a week and after the first week of May. The
contribution from Bangladesh was sporadic comparing to other regions. Highest contribution
from this Bangladesh region was observed after the first week of June. Pakistan also contributed
for the CO loading significantly. Others region as mentioned in the figure covered the regions
like Afghanistan, Middle east, West Asia, East Asia, Africa and Bhutan. Contributions from
these regions were less than 5%. Contribution from China was not evident till the first week of
June where a specific air mass arrival shows contribution reaching up to 25% of total CO
loading.
A sensitivity analysis was performed for emission uncertainty in the model grid containing
Lumbini. Lumbini and surrounding regions in the recent years has seen significant rise in urban
activities and industrial activity and related emissions which may not be accurately reflected in
the HTAPv2 emissions inventory. A month long simulation was carried out with emissions from
Lumbini and the surrounding four grids off and another simulation with Lumbini and
surrounding four grid's emissions scaled 5 times the amount from HTAPv2 emissions inventory.
The results are shown in Figure 12 (C) as percentage increase or decrease compared to model
results using the current HTAPv2 emissions inventory. The black line shows concentration as
100% for the current HTAPv2 emissions inventory. Despite making Lumbini and the
surrounding grids emissions zero, model calculation shows pollutant concentration on average is
still about 78% of the original value indicating dominance of background and regional sources
compared to local source in the model. Increasing emissions 5 times for the Lumbini and
surrounding four grids only increases the concentration on average by 151%. Thus uncertainty in
emissions are not a local uncertainty for Lumbini rather for the whole region which needs to be
better understood for improving model performance against observations at Lumbini.
**3.4      Contribution of aerosol composition to local air quality as identified by the model**
The chemical composition of $PM_{2.5}$ obtained from the model is shown in Figure 13.
Carbonaceous aerosols and sulfate pollutants contributed two-third fraction of the fine mode
particulate matter ($PM_{2.5}$). Organic carbon (OC) was found as the main constituent of the $PM_{2.5}$



contributing ~ 45% to PM$_{2.5}$. For Lumbini, the contribution of modeled BC to PM$_{2.5}$ was ~ 10%
similar to the observed (9.2%) fraction of BC to PM$_{2.5}$. Recent study conducted over nearby IGP
site, Kanpur (Ram and Sarin, 2011) found the average share of OC and EC in PM$_{2.5}$ to be ~45%
and ~5% respectively which is close to the values obtained by our model based calculation.
Natural aerosols mainly wind-blown mineral dust was ~ 25% of the fine mode PM in Lumbini.
Highest loading of dust is observed during the late dry period to early monsoon season in South
Asian region (Adhikary et al., 2007). Sulfate contributed for ~ 20% share of the PM$_{2.5}$ over
Lumbini. Although the post monsoon season observed highest concentration of sulfate in South
Asian region, elevated concentration are observed even during the April over Ganges Valley
(Adhikary et al., 2007). As expected, very minimal contribution from sea salts (less than 1%)
was observed at Lumbini.

### 3.5 Does fossil fuel or biomass influence the Lumbini air?

The aerosol spectral absorption is used to gain insight into nature and potential source of black
carbon. This method enables to analyze the contributions of fossil fuel combustion and biomass
burning contributions to the observed BC concentration (Kirchstetter et al., 2004). Besides BC,
other light absorbing (in the UV region) aerosols are also produced in course of combustion,
collectively termed as organic aerosols (often also called brown carbon or BrC) (Andreae and
Gelencsér, 2006). Figure 14 shows the comparison of normalized light absorption as function of
the wavelength for BC observed at Lumbini during cooking and non-cooking hours. Our results
are compared with the published data of Kirchstetter et al. (2004) and that observed over a
village center site of Project Surya in the IGP (Praveen et al., 2012) (figure not shown). We
discuss light absorption data from two distinct times of the day. The main reason behind using
data during 07:00-08:00 h and 16:00-17:00 h is these periods represent highest and lowest
ambient concentration (Fig. 7). Also these period represent cooking and non-cooking or high and
low vehicular movement hours (Praveen et al., 2012). To understand the influence of biomass
and fossil fuel we plotted normalized aerosol absorption at 700 nm wavelength for complete
aethalometer measured wavelengths in Fig. 14. Kirchstetter et al. (2004) reported OC absorption
efficiency at 700 nm to be zero. Thus we normalized measured absorption spectrum by 700 nm
wavelength absorption. Since aethalometer does not provide 700 nm wavelength absorption
values, we used methodology followed by Praveen et al. (2012). Our results show that the





normalized absorption for biomass burning aerosol is ~3 times higher at 370 nm compared to
that at 700 nm whereas fossil fuel absorption is about 2.6 times higher at the same wavelength.
The normalized curve obtained during both cooking and non-cooking period lies in between the
standard curve of Kirchstetter et al. (2004). The curve during the prime cooking time is much
close to the biomass curve of published data (including that during the cooking period over the
village center site of Project Surya) whereas that during non-cooking time (afternoon period) is
inclined towards the fossil fuel curve. Similar result was also observed over the Project Surya
village in the IGP region (Praveen et al., 2012; Rehman et al., 2011). This clearly indicates there
is contribution of both sources: biomass as well as fossil fuel on the observed BC concentration
over Lumbini.
In order to identify fractional contribution of biomass burning and fossil fuel combustion to
observed BC aerosol, we adopted the method described by Sandradewi et al. (2008). Wavelength
dependence of aerosol absorption coefficient ($b_{abs}$) is proportional to $\lambda^{-\alpha}$ where $\lambda$ is the
wavelength and $\alpha$ is the absorption Ångstrom exponent. The $\alpha$ values ranges from 0.9-2.2 for
fresh wood smoke aerosol (Day et al., 2006) and between 0.8-1.1 for traffic or diesel soot
(references in Sandradewi et al. (2008)). We have taken $\alpha$ value of 1.86 for biomass burning and
1.1 for fossil fuel burning as suggested by previous literature (Sandradewi et al., 2008). Figure
15 shows diurnal variation of the biomass burning BC. Minimum contribution of biomass
burning to total BC concentration was observed during 04:00-06:00 local time (only about 30%
of the total BC). As the cooking activities start in morning, the contribution of biomass BC starts
to increase and reaches about 50%. Similar pattern was repeated during evening cooking hours.
Only during these two cooking periods, fossil fuel fraction BC was lower. Otherwise it remained
significantly higher than biomass burning BC throughout the day. On average, ~40% of BC was
from biomass burning whereas remaining 60% was contributed by fossil fuel combustion during
our measurement period. Interestingly, this is the opposite of the contributions that were
concluded by Lawrence and Lelieveld (2010). Lawrence and Lelieveld (2010) concluded that
~60% BC from biomass versus ~40% fossil fuel, based on a review of numerous previous
studies to be likely for the outflow from Southern Asia during the winter monsoon. When we
compared observed Ångstrom exponent with Praveen et al. (2012), we noticed that Lumbini
values were lower than Project Surya Village center site. This implies Surya village center had





553 higher biomass fraction, also it was observed absorption Ångstrom exponent exceeded 1.86

554 during cooking hours which indicates 100% biomass contribution. The difference is attributed to

555 the fact that Lumbini sampling site is not a residential site like Surya village which can capture

556 cooking influence efficiently. Further Lumbini sampling site is surrounded by commercial

557 activities such as a local bus park, hotels, office buildings and industries and brick kilns slightly

558 further away. Although the reason for this difference is not clear, it is an indication of the

559 important role of diesel and coal emissions in the Lumbini and upwind regions.

560 **4 Conclusions**

561 Our measurements, a first for the Lumbini area, have shown very high pollution concentration at

562 Lumbini. Black carbon (BC), carbon monoxide (CO), ozone ($O_3$) and particulate matter ($PM_{10}$,

563 $PM_{2.5}$ and PM1) were measured during the pre-monsoon of 2013 as a regional site of the *SusKat-*

564 *ABC campaign*. Average pollutant concentrations during the monitoring period were found to be:

565 BC: $4.9\pm3.8$ µg m$^{-3}$; CO: $344.1\pm160.3$ ppbv; $O_3$: $46.6\pm20.3$ ppbv; $PM_{10}$: $128.8\pm91.9$ µg m$^{-3}$ and

566 $PM_{2.5}$: $53.14\pm35.1$ µg m$^{-3}$ which is comparable with other urban sites like Kanpur and Delhi. The

567 diurnal variation of the pollutants is similar to that of any urban location, with peaks during

568 morning and evening. However, our results show higher evening concentration compared to

569 morning concentration values. During our measurement period, air quality in Lumbini was

570 influenced by regional forest fires as shown by model and satellite data analysis. A regional

571 chemical transport model, WRF-STEM was used to interpret observations. Inter-comparison of

572 WRF-STEM model outputs with observations showed that the model underestimated the

573 observed pollutant concentrations by a factor of 1.5 to 5. Nonetheless, WRF-STEM model was

574 able to simulate the synoptic variability of observed pollutants. Model uncertainties are attributed

575 mostly to uncertainties in meteorology and regional emissions. Region-tagged CO as air-mass

576 tracers are employed in STEM to understand the source region influencing Lumbini. Our

577 analysis shows that the adjacent regions; mostly the Ganges valley, other parts of India and

578 Nepal accounted for the highest contribution to pollutant concentration in the Lumbini.

579 Anthropogenic pollutants in $PM_{2.5}$ were dominant, with OC and BC contributing ~ 45% and

580 ~10%, respectively while sulfate aerosol contributed to 20%, whereas natural pollutants like

581 mineral dust contributed ~ 25%. The normalized light absorption curve clearly indicated the





contribution to BC in Lumbini from both sources: biomass as well as fossil fuel. On average,
~40% BC was found to be from the biomass burning and ~60% from fossil fuel burning.
Various improvements and extensions would be possible in future studies. More reliable
functioning of the AWS (temperature and RH sensor, rain gauge) would have allowed more in-
depth analysis of the relationship between meteorological parameters and pollutants
concentration. Continuous measurements of air pollutants throughout the year would allow for
annual and seasonal variation study. Improvements in the model are much needed in its ability to
simulate observed meteorology. Significant uncertainty lies with regional emissions inventory
and emissions from open burning.
There is a clear need for setting up of a continuous air quality monitoring station at Lumbini
(UNESCO World Heritage Site) and the surrounding regions for long-term air quality
monitoring. In order to fully safeguard the valuable world heritage properties as well as public
health and agro-ecosystems in the region from impacts of air pollution, development activities
within the Kenzo Tange Master Plan Area and Lumbini Protected Zone (LPZ) need to go
through a rigorous environmental impact assessment (EIA) and heritage impact assessment
(HIA) in accordance with the decisions of the UNESCO World Heritage Committee.

**Data availability**
The data used for this manuscript can be obtained by sending an email to the corresponding
authors and/or to IASS (Maheswar.Rupakheti@iass.potsdam.de) and/or to ICIMOD
(arnico.panday@icimod.org). Modeling code can be obtained from B. Adhikary
(Bhupesh.adhikary@icimod.org).

**Authors' contributions**
M.R. designed the experiment. D.R. and K.S.M conducted the field observations. B.A. ran the
WRF-STEM model. D.R., B.A., P.S.P., M.R. and S.K. conducted the data analysis, and D.R.
prepared the manuscript with inputs from all coauthors.





**Acknowledgements**

This study was partly supported by the Institute for Advanced Sustainability Studies (IASS), Germany, the International Centre for Integrated Mountain Development (ICIMOD), and the National Natural Science Foundation of China (41121001, 41225002), and the Strategic Priority Research Program (B) of the Chinese Academy of Sciences (XDB03030504). D. Rupakheti would like to acknowledge the CAS-TWAS President PhD Fellowship program. ICIMOD authors would like to acknowledge that this study was partially supported by core funds of ICIMOD contributed by the governments of Afghanistan, Australia, Austria, Bangladesh, Bhutan, China, India, Myanmar, Nepal, Norway, Pakistan, Switzerland, and the United Kingdom. The views and interpretations in this publication are those of the authors and are not necessarily attributable to the institutions they are associated with. We thank B. Kathayat, B.R. Bhatta, and Venerable Vivekananda and his colleagues (Panditarama Lumbini International Vipassana Meditation Center) for providing logistical support which was vital in setting up and running the site. We also thank C. Cüppers and M. Pahlke of the Lumbini International Research Institute (LIRI) for proving the space and power to run the instruments at the LIRI premises. Satellite data providers (MODIS, AIRS, OMI) are also equally acknowledged.



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



**Table 1.** Summary of instruments deployed during monitoring in Lumbini

| Instrument (Model) | Manufacturer | Parameters | Inlet/sensor height (above ground) | Sampling interval | Sampled period |
|---|---|---|---|---|---|
| Environmental Dust monitor (EDM 164) | GRIMM Aerosol Technik, Germany | $PM_{10}$, $PM_{2.5}$, $PM_1$ | 5 m | 5 min | 01/04-15/06 |
| Aethalometer (AE42) | Magee Scientific, USA | Aerosol light absorption at seven wavelengths, and BC concentration | 3 m | 5 min | 01/04-05/06 |
| CO analyzer (48i) | Thermo Scientific, USA | CO concentration | 3 m | 1 min | 01/04-15/06 |
| $O_3$ analyzer (49i) | Thermo Scientific, USA | $O_3$ concentration | 3 m | 1 min | 01/04-15/06 |
| Automatic Weather Station (AWS) | Campbell Scientific, UK | T, RH, WS, WD, Global Radiation, Precipitation | 12 m | 1 min | 01/04-15/06 |








**Table 2.** Comparison of PM$_{2.5}$, BC, CO and O$_3$ concentrations at Lumbini with those at other sites in South Asia

| Sites | Characteristics | Measurement period | PM$_{2.5}$ (μg m$^{-3}$) | BC (μg/m$^3$) | CO (ppbv) | O$_3$ (ppbv) | References |
|---|---|---|---|---|---|---|---|
| Lumbini, Nepal | Semi-urban | Pre-monsoon, 2013 | 53.1±35.1 | 4.9±3.8 | 344.1±160.3 | 46.6±20.3 | This study |
| Kathmandu, Nepal | Urban | Pre-monsoon, 2013 | - | 14.5±10 | - | 38.0±25.6 | (Putero et al., 2015) |
| Mt. Everest, Nepal | Remote | Pre-monsoon | - | 0.4±0.4 | - | 61.3±7.7 | (Marinoni et al., 2013) |
| Delhi, India | Urban | Pre-monsoon (night-time) | 82.3±50.5 | 7.70±7.25 | 1800±890 | - | (Bisht et al., 2015) |
| Kanpur, India | Urban | June 2009-May 2013, April-June | - | 2.1±0.9 | 721±403 | 27.9±17.8 | (Gaur et al., 2014) (Ram et al., 2010) |
| Mohali, India | Semi-urban | May, 2012 | 104±80.3 | - | 566.7±239.2 | 57.8±25.4 | (Sinha et al., 2014) |
| Mt. Abu, India | Remote | Jan 1993-Dec 2000, pre-monsoon | - | 0.7±0.14 | 131±36 | 39.9±10.8 | (Naja et al., 2003) (Das and Jayaraman, 2011) |





**Table 3**. Inter-comparison of observed and model simulated hourly average concentrations of air
pollutants during the measurement campaign period. Unit: PM and BC in µg/m$^3$ and CO in ppbv.

| Pollutants | Observed (mean and range) | Modeled (mean and range) | Ratio of mean (observed/modeled) |
|:---:|:---:|:---:|:---:|
| $PM_{10}$ | 128.8 (10.5-604.0) | 25.4 (2.1-68.8) | 5 |
| $PM_{2.5}$ | 53.1 (6.1-272.2) | 17.3 (1.9-48.3) | 3 |
| CO | 344.1(124.9-1429.7) | 255.7 (72.2-613.1) | 1.4 |
| BC | 4.9 (0.3-29.9) | 1.8 (0.4-3.7) | 2.7 |







**Figures**

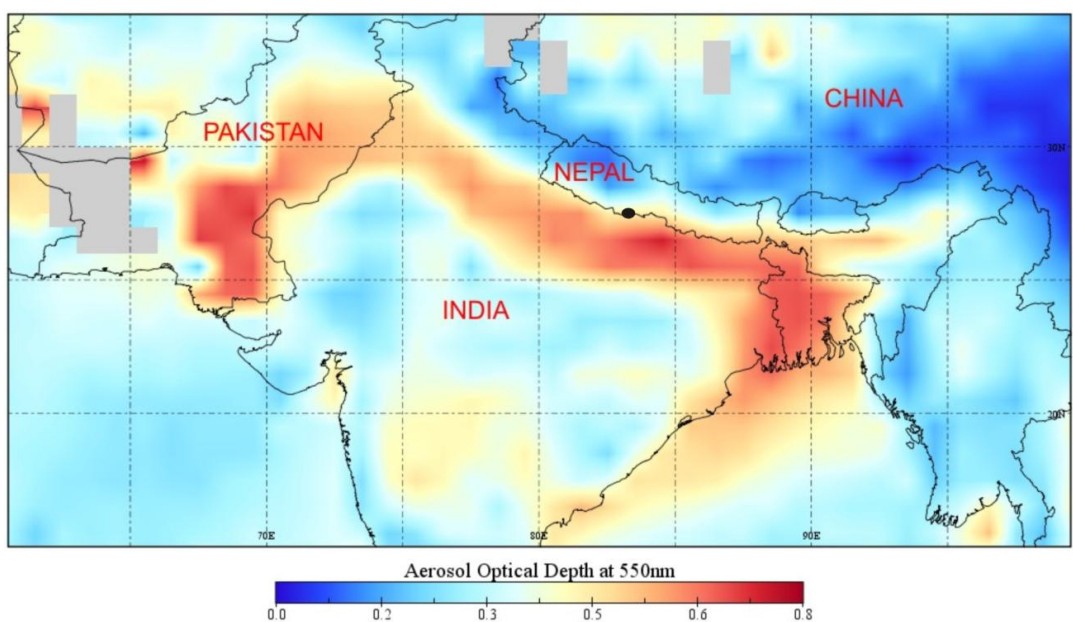


**Figure 1.** Aerosol optical depth in South Asia acquired with the MODIS instrument aboard
TERRA satellite averaged over the winter and pre-monsoon season (December 2012-June 2013).
High aerosol loading can be seen over the entire Ingo-Gangetic Plains (IGP). An aerosol hotspot
south of Lumbini (small black mark nearby the border of Nepal with India) is clearly visible.
Light grey color used in the figure represents the absence of data.



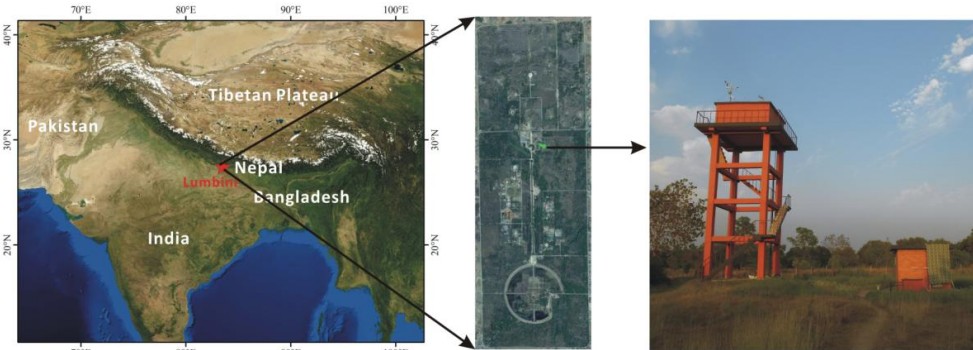


**Figure 2.** Location of sampling site in Lumbini in southern Nepal (left panel). The middle panel
shows the Kenzo Tange Master Plan Area of Lumbini while the right panel shows the sampling
tower in the Lumbini Master Plan Area.




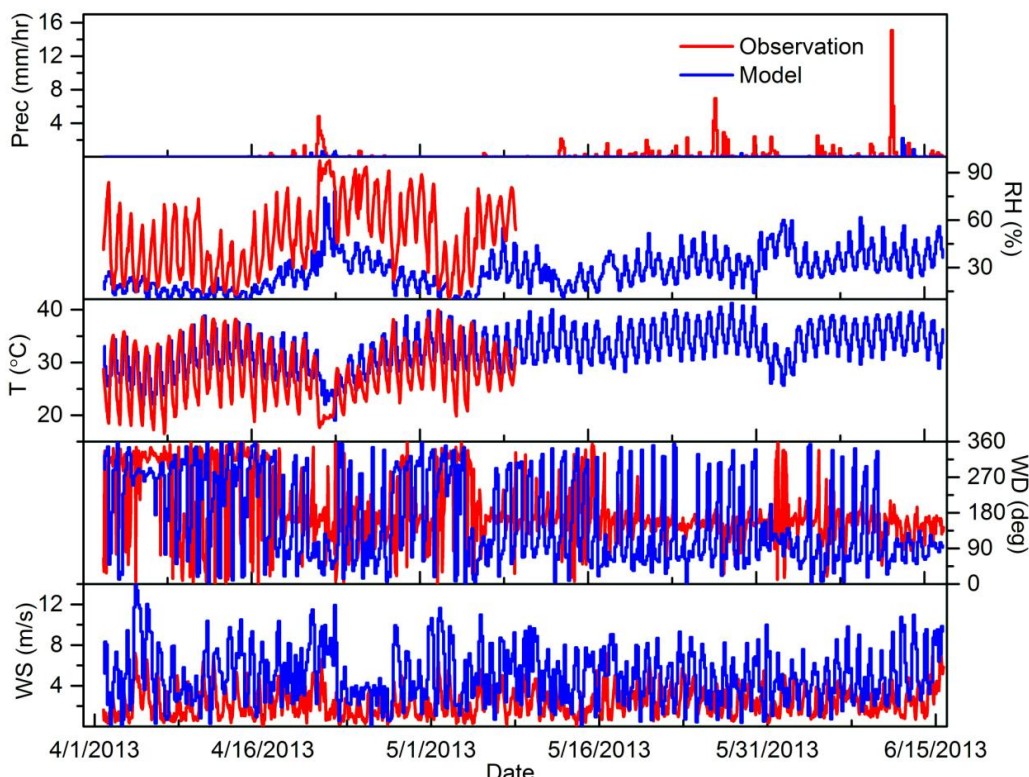


**Figure 3.** Time series of hourly average observed (red line) and model estimated (blue line) meteorological parameters at Lumbini, Nepal for the entire sampling period from 1 April to 15 June 2013.









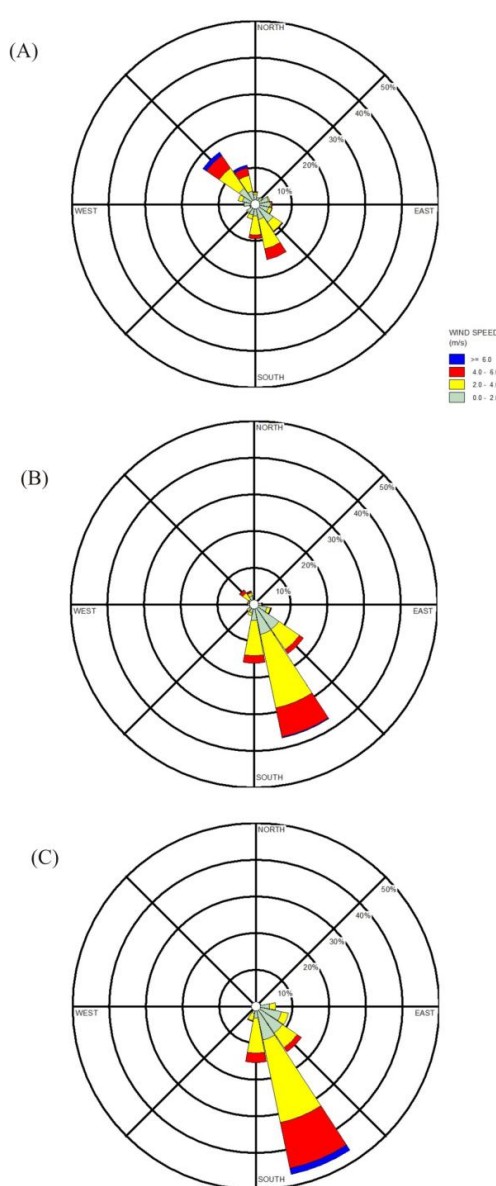


**Figure 4**. Wind rose of wind speed and wind direction observed at Lumbini during the month of
(A) April, (B) May, and (C) (1st-15th) June 2013.

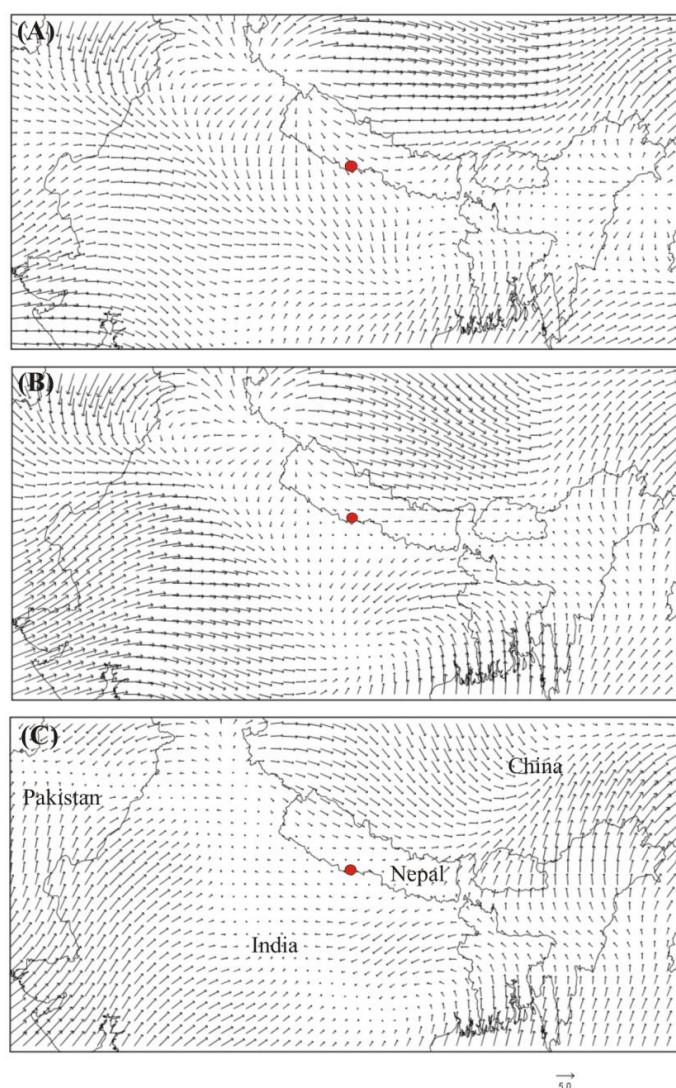


**Figure 5.** Monthly synoptic surface winds for the month of (A) April, (B) May and (C) June

2013, based on NCEP/NCAR reanalysis data. Orientations of arrows in the figures refer to wind

direction whereas the length of arrows represents the magnitude of wind speed (m/s). Red dot in

the map represents the location of Lumbini.

840



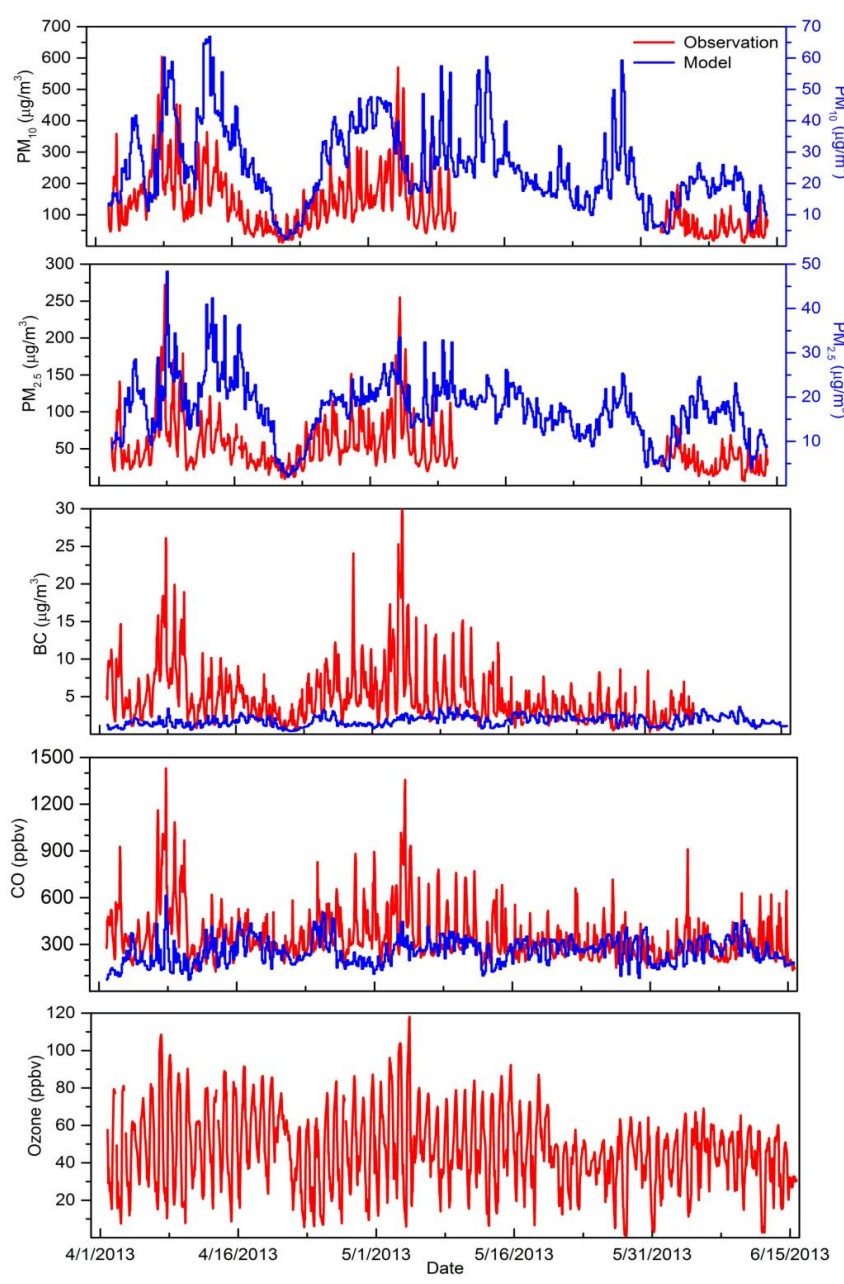

841

**Figure 6.** Time series of the observed (red line) and model estimated (blue line) hourly average

concentrations of PM$_{10}$, PM$_{2.5}$, BC, CO and O$_3$ at Lumbini, Nepal for the entire sampling period

from 1 April to 15 June 2013. Model estimated O$_3$ was not available.

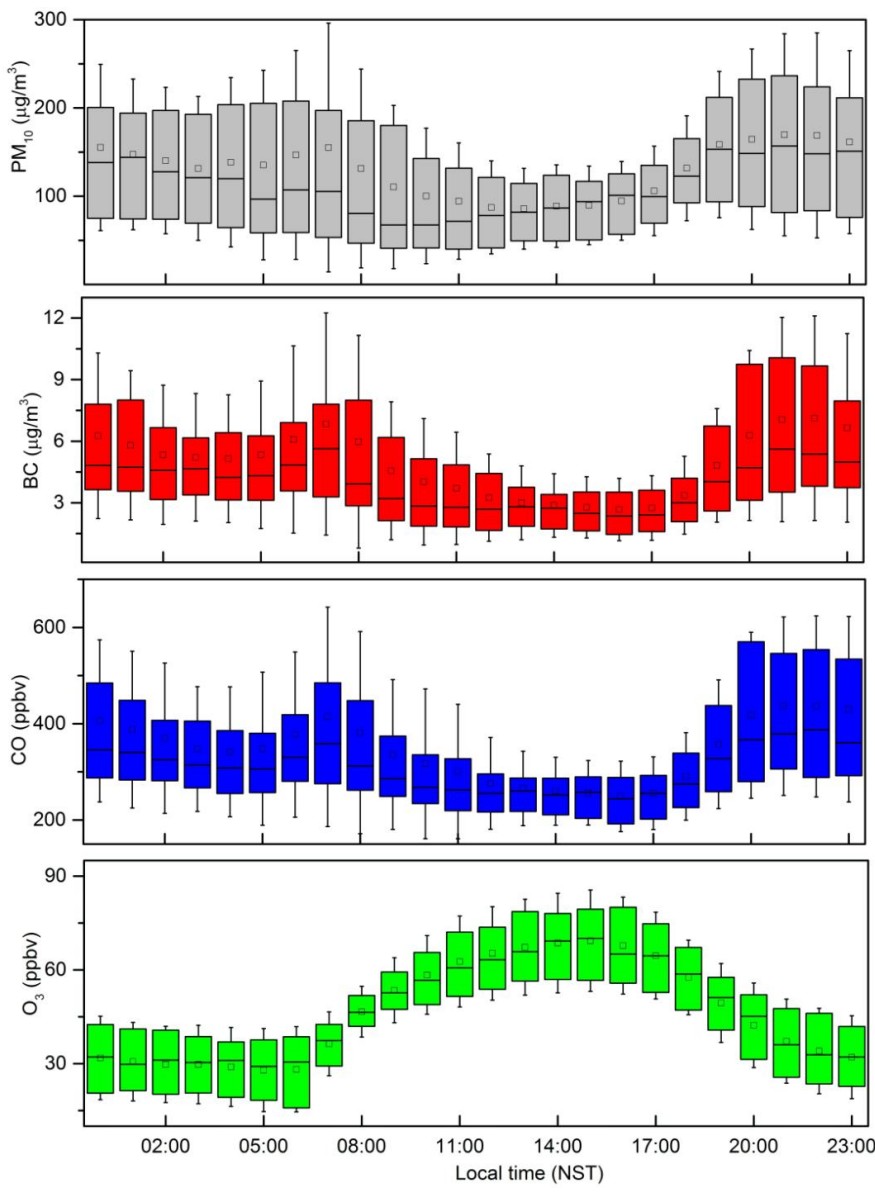

845

**Figure 7.** Diurnal variations of hourly average ambient concentrations of PM$_{10}$, BC, CO and O$_3$

at Lumbini during the monitoring period (1 April -15 June 2013). In each box, lower and upper

boundary of the box represents 25[th] and 75[th] percentile respectively, top and bottom of the

whisker represents 90[th] and 10[th] percentile respectively, the mid-line represents median, and the

square mark represents the mean for each hour.





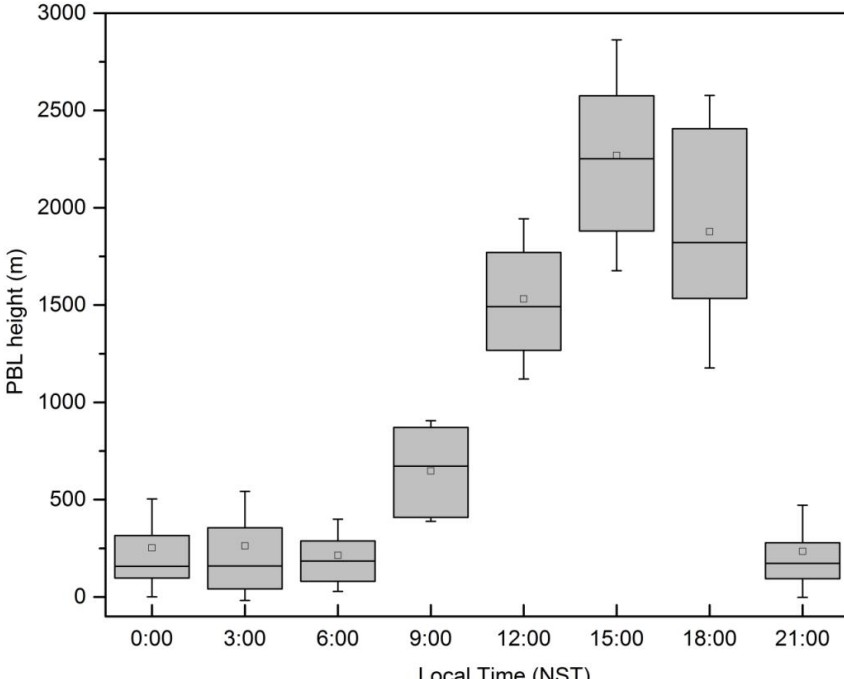

851

**Figure 8.** Diurnal variation of the planetary boundary layer (PBL) height at Lumbini obtained
for every three hours of each day from the WRF-STEM model for the sampling period. The
square mark in each box represents the mean PBL height, bottom and top of the box represents
$25^{th}$ and $75^{th}$ percentile, top and bottom of the whisker represents $90^{th}$ and $10^{th}$ percentile
respectively.





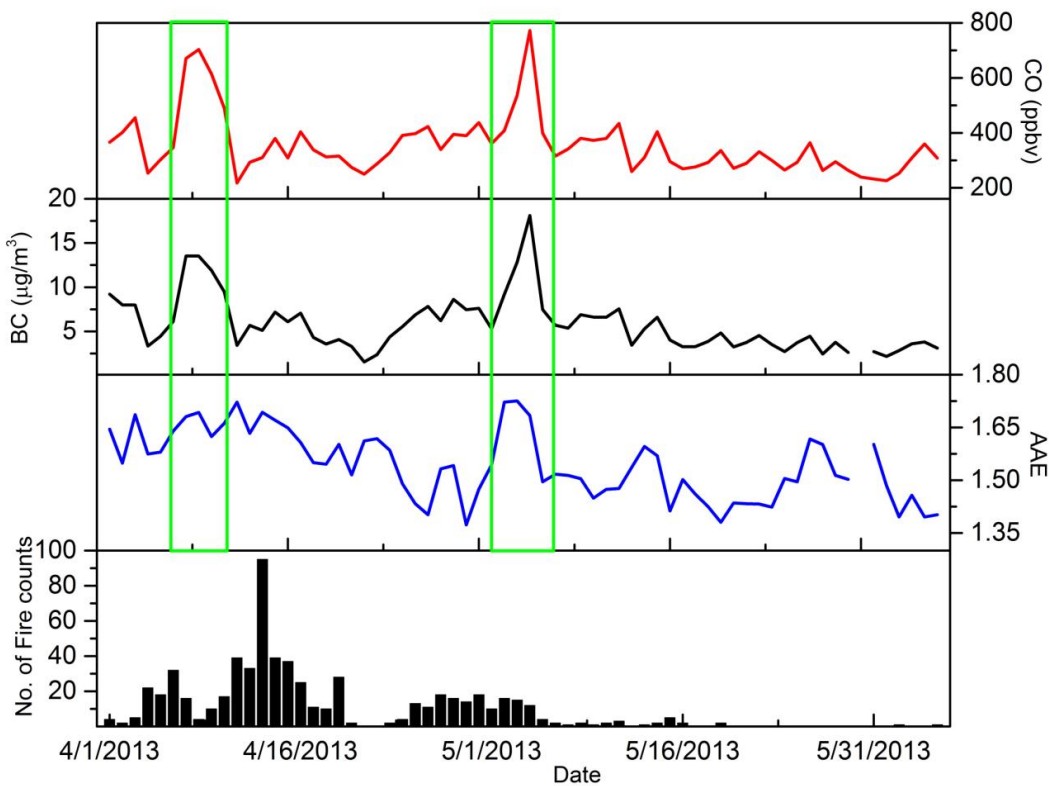

857

**Figure 9**. Time series of daily average CO, BC concentration, absorption Ångstrom exponent

(AAE), along with fire counts acquired with the MODIS instrument onboard TERRA satellite for

a 200x200 km grid centered at Lumbini. Two rectangular green boxes represent two episodes

with high peaks in CO and BC concentrations.






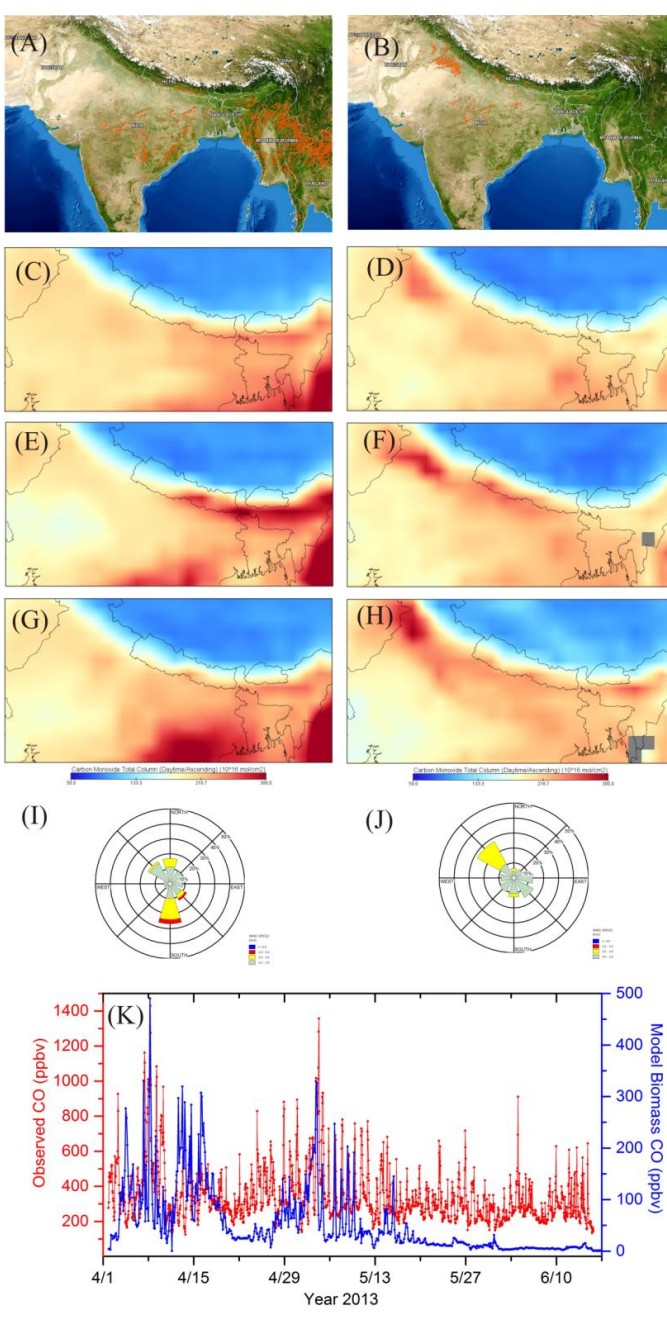


**Figure 10.** Active fire hotspots in the region acquired with the MODIS instrument on TERRA
satellite during (A) Event-I (7-9 April) and (B) Event-II (3-4 May). CO emissions, acquired with



AIRS satellite, in the region 2 days before (3-5 April), during (7-9 April) and 2 days after (10-12
April) the Event-I are shown in panels (C), (E) and (G), respectively while panels (D), (F) and
(H) show CO emissions 2 days before (1-2 May), during (3-4 May) and 2 days after (5-6 May)
the Event-II. Panels (I) and (J) represent the average wind rose plot of observed wind direction
and wind speed during Event I and II, respectively. (K) Observed CO versus Model open
burning CO illustrating contribution of forest fires during peak CO loading.





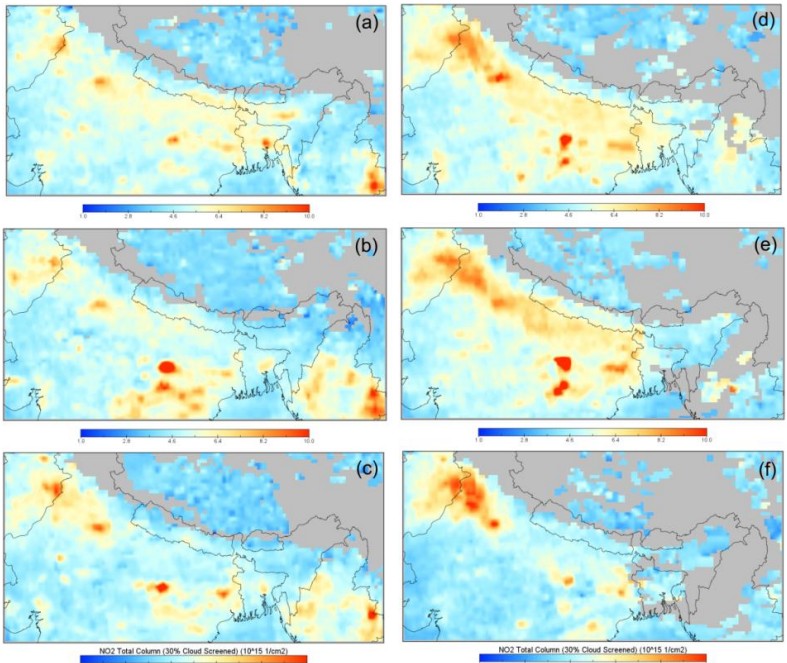


**Figure 11.** NO$_2$ total column obtained with OMI satellite over the region (a) before, (b) during,
and (c) after the Event- I. The panels (d), (e), (f) show NO$_2$ total column before, during and after
the Event- II.





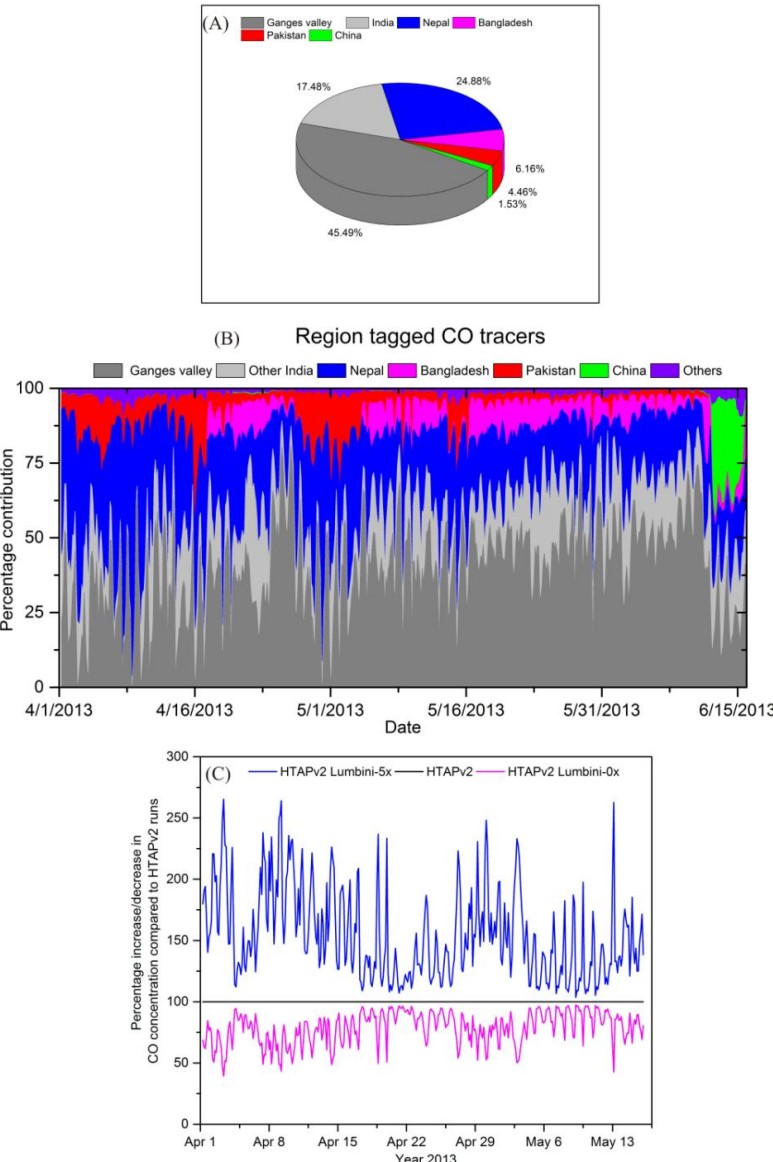


**Figure 12.** (A) WRF-STEM model estimated contributions of various source regions to average CO concentration in Lumbini for the sampling period, (B) time series of region tagged CO tracer during the whole measurement period using HTAP emission inventory and (C) Figure showing percentage increase/decrease in CO concentration with different emissions scenario.






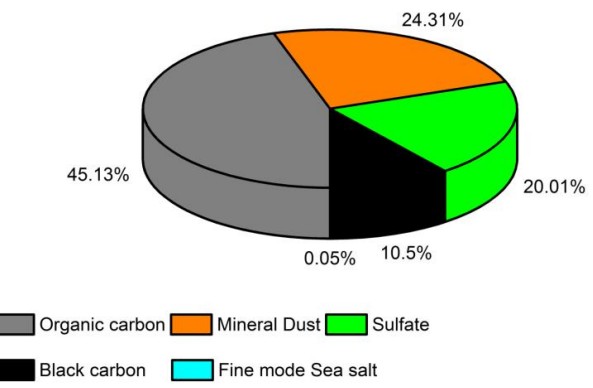


**Figure 13.** WRF-STEM model estimated PM$_{2.5}$ chemical composition at Lumbini for pre-
monsoon season 2013





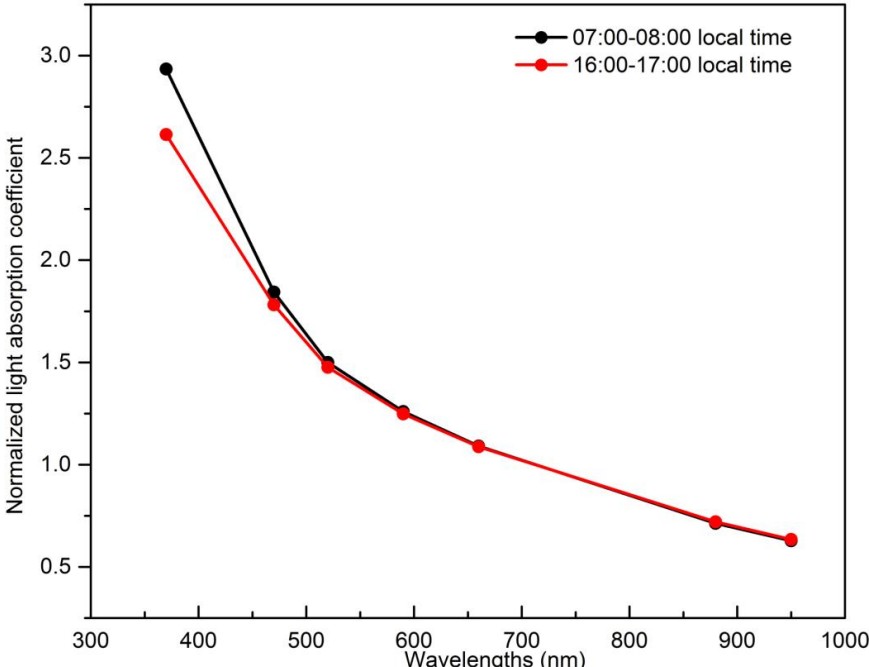


**Figure 14.** Normalized light absorption coefficients during cooking (07:00-08:00) and non-cooking (16:00-17:00) period based on diurnal variation of BC at Lumbini during the sampling period in premonsoon season 2013.







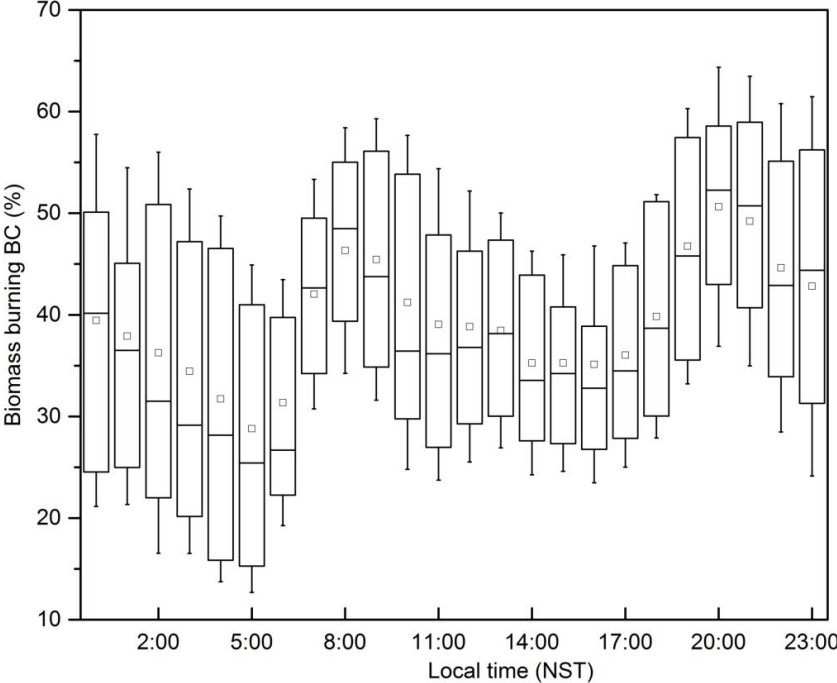


**Figure 15.** Diurnal variation of the fractional contribution of biomass burning to ambient BC concentration at Lumbini for the measurement period. In each box, lower and upper boundary of the box represent 25[th] and 75[th] percentile, respectively, top and bottom of the whisker represents 90[th] and 10[th] percentile, respectively. The mid-line in each box represents median while the square mark represents the mean for each hour.