# Peer review of "Pre-monsoon air quality over Lumbini, a world heritage site along the Himalayan foothills"

_Atmospheric Chemistry and Physics, 2016_

## Referee Comment (RC1) · Anonymous Referee #1 · 21 Jul 2016

In this paper, authors present measured ambient PM, BC, CO, and $O_3$ concentrations in Lumbini during an intensive measurement campaign from April – June, 2013. They also conducted a regional WRF-STEM modeling to simulate the meteorology and air pollutant concentrations, as well as to examine the aerosol chemical composition. The authors conclude that there is high pollution in Lumbini and that a network of long-term air quality monitoring stations is needed in the greater Lumbini region. I agree with the authors that it is important to collect observational data in this area and the set of observations presented in the paper is extremely useful for understanding the magnitude of the air pollution problem and the potential sources. However, the language is very vague and the scientific discussion is limited. I find that improvements are essential before the manuscript can be published in ACP. I list some of the concerns below.

I am unsure if what we are most interested in is the comparison of Lumbini measurements with those of other cities. For example, they state that "BC observed at Lumbini was higher by a factor of ~6 and ~4.5 compared to that at Mt. Abu, India and near the base of Mt. Everest, Nepal, respectively (l. 323-326)" but how do we know what to make of these comparisons. What do we learn from these comparisons? To me, it is logical that Lumbini has higher BC concentrations compared to those remote places. Similarly, I do not find interesting that the ozone concentrations were higher at Lumbini than in the Mt. Everest (l. 332). I would rather be more interested to know how the monthly concentrations change and when the highest and lowest concentration levels were and if there was any difference in the three months or over time among the species. Also, examining the period when the model is able to reproduce observations and contrasting that to the times when the model fails would be a good way to make use of both measurements and the model. Such assessment should also provide a good basis for what needs to be improved in the model. I find the argument that the authors put forward on l. 261-262 that the "[D]iscrepancy on model results might have occurred due to various factors inherently uncertain in a weather model" to be hand-waving and not really helpful. With this data set, they should be able to understand the discrepancy between the model and the observations a little better.

Why is PM1 concentration not discussed in the study (l. 282-283)? If it is discussed elsewhere, please mention it. If there was a problem with the data, then I think this should not be included in this manuscript at all. If there was no problem with the data, I think that can provide an additional insight into the measurements and is worth exploring more than just the average concentrations mentioned in l. 281.

I am quite confused about WRF-STEM model simulations. Authors state: "A comparison of model calculated average concentration along with the minimum and maximum concentrations of various pollutants (with observation) is shown in Table 3 (l. 340-342)." However, right after this sentence, they write that "[T]he model based concentrations used here are instantaneous values for every third hour of the day (l. 342-343)." Can authors clarify which one that is and if the latter, why did they use the instantaneous values?

There are many places where authors state in a very qualitative manner, which obviously is not helpful for the reader to understand the issues being discussed. I list some of the sentences here:

1. l. 298-299 "BC to CO ratio in Lumbini was found to be different from that observed at other urban and rural sites and those affected by forest fire/biomass burning." What was the ratio observed at Lumbini and at other places? What can we infer from this? What is the criterion for "different"?

2. l. 299-302 "a suburban site, Pantnagar, in IGP also observed similar BC to CO ratio." What value is considered "similar" and how is that determined? What do we learn from this?

3. l. 318-321 "$PM_{2.5}$ concentration in Lumbini have been found to be lower than the megacity like Delhi and north-western IGP regions due to higher level of emissions over those regions." How did they come up with this conclusion? I do not see any comparison of emissions, especially at the sector level. Also, I understood that changing emissions in Lumbini and surrounding regions did not lead to a large concentration difference in the model when they conducted a sensitivity analysis (l. 474-488). Doesn't this conflict with what is argued here?

4. l. 321-323 "BC concentrations observed in Lumbini during pre-monsoon season was lower than the urban Asian cities like Kathmandu and Delhi, slightly higher than in Kanpur but high compared to the remote locations in the region." Are the authors comparing the measurements during the same period between cities? What does "slightly higher" and "high" mean? What is the definition of these? More importantly, what do we learn from this?

5. l. 355-359 "STEM model performance can be significantly improved via better constraining anthropogenic emissions inventory, emissions of open biomass burning and improvements in meteorological output from WRF amongst many other uncertainties inherent in regional chemical transport model." How did they get to this conclusion?

6. l. 526-529 "The curve during the prime cooking time is much close to the biomass curve of published data whereas that during non-cooking time is inclined towards the fossil fuel curve." How is "much close" determined, as well as "inclined"?

For the two events when authors found an elevated BC and CO concentrations, what were the PM and $O_3$ levels? Did they find an elevated PM on any other days? Did they find an elevated potassium level during those days? I think that focusing on the analysis of these two events and clearly explaining the details of the regional contribution assessment presented in the manuscript would definitely strengthen the paper. The regional contribution assessment could be also extended by quantifying the monthly differences and also considering other species. This then could be linked to the chemical composition to assess if the regional contribution has anything to do with the chemical composition difference that they can potentially see in different months.

Minor comments:
1. rain guage → rain gauge (l. 239)
2. I'm not sure if the authors really meant the way they wrote the sentence: "But, to our expectation, we could not observe any significant influence of forest fires within the specified grid (l. 419-420)." Did the authors really expect that they would not be able to observe influence? Or is this a typo?
3. Others region → other regions (l. 469)

---

## Referee Comment (RC2) · Anonymous Referee #2 · 7 Nov 2016

General comments

The paper reports for the first time in Lumbini, Nepal, BC, CO, O3 and PM data from a 3-month experiment in one site. The motivation is to understand air quality in Lumbini, but this objective sounds oversized in regard of the limited duration of the experiment. By the way, no scientific question is set and the methodology presents weak points. Data are new but rather few. No chemical speciation is provided to complete the data of species monitored online. Moreover, those online data could have been further treated: by using ratios (e.g., BC/CO, K/BC, PM1/PM2.5) and the aethalometer model to take full benefit from the BC spectral dependence. The use of modeling is useful to study the synoptic variability of pollutants, but appears highly questionable to simulate the chemical components, given the poor emission data used. A shorter manuscript, attempting to better understand for instance the source effects of the major emission

points affecting Lumbini, using data only, not modelling, could be considered.

Specific comments

22 Abstract: Objectives and/or a scientific question need to be clearly stated. 178 Was any cut-off applied on the BC sampling line, or was it bulk BC? 282 It is a pity that PM1 was not considered. The variations of the PM1-to-PM2.5 ratio would possibly provide interesting information about source profiles. 296 Both BC and CO are from incomplete combustion processes but the ratio BC/CO is often specific to the different processes. A plot of the variations of BC/CO in time could be more relevant than BC and CO separately. 414 BC and CO originate from biomass burning as well as from any other fuel and combustion types, as mentioned earlier (line 296). Thus this sentence does not justify the use of BC and CO. Instead, BC/CO could help for source discrimination. 417 Potassium is a biomass burning tracer when the fine fraction is considered. As it can have other sources, it is rather examined as K/BC. 563 Remove PM1

Technical corrections

86 "to be the most" 150 border 182 "A similar value" 211 latitudes and longitudes, why "s" 232 "viz." what is the meaning? 239 "gauge" 283 "24-hour" 369 "the emission inventory shows" 388 "15:00" remove "h" 487 "is not" 516 "these periods"
* * *

---

## Referee Comment (RC3) · Anonymous Referee #3 · 9 Dec 2016

The authors made a good attempt to conduct the monitoring and modeling studies for the selected air pollutants over the study area. However, the current MS should be further improved before it can be reconsidered for the publication in ACP. Major comments: 1) It is not clear what hypothesis the authors want to test in this study hence the content is quite diluted and is difficult to follow the MS. 2) The linkage between the modeling and monitoring parts appear to be quite weak. How the results of both parts supported each other to reach the study objectives (and what are these?)?. If both monitoring and modeling results are to be incorporated then the purpose/research question should be clearly defined from the beginning. 3) In my opinion, it would be more interesting if the authors make better attempt to analyze the monitoring data (including also PM1, O3 etc.) in relation to the sources and meteorology, etc. rather than to loosely cover all the activities/results as presented in this version. 4) The methodology for the modeling part should be described in detail, especially the emission input data. The authors claimed in Line 436 that both modeling and monitoring results showed CO peaks during the biomass burning events but not indicated if and how the emissions from these 2 events were also included in the emission input data. Minor comments: 1) The description of monitoring instrument (2.2) is lengthy and could be moved to SI. 2) Too many qualitative statements in the MS.

---

## Author Comment (AC1) · 10 Feb 2017

February 10th, 2017

Dear Editor,

At first, we would like to thank you for serving as the editor for our work.

We would like to inform you that we have revised the manuscript significantly based on the comments by three reviewers. We believe that the comments have certainly helped improve the quality of our analysis and interpretations. We have shortened various sections, and also inserted new sections in order to incorporate reviewers' comments and hence strengthen our analysis and discussion. Sections on ratios of PM fractions, BC/CO ratios, and influence of meteorology on pollution concentrations have been inserted whereas section on model based aerosol composition has been taken out following the reviewer's suggestions. Moreover, new figures have also been included to explain the findings from our work. Finally, the abstract and conclusion section have been rewritten to reflect the updated findings.

Sincerely Yours,

Dipesh Rupakheti and Prof. Shichang Kang, on behalf of all coauthors

**Pre-monsoon air quality over Lumbini, a world heritage site along the Himalayan foothills**

by D. Rupakheti et al., 2016 (ACPD)

We would like to thank the reviewers for their constructive comments and suggestions. Please find the reviewer's comments in black and our replies in blue. The changes in the revised manuscript are colored in red.

**REVIEWER-1**

In this paper, authors present measured ambient PM, BC, CO and O3 concentrations in Lumbini during an intensive measurement campaign from April-June 2013. They also conducted a regional WRF-STEM modeling to simulate the meteorology and air pollutant concentrations, as well as to examine the aerosol chemical composition. The authors conclude that there is high pollution in Lumbini and that a network of long-term air quality monitoring stations is needed in the greater Lumbini region. I agree with the authors that it is important to collect observational data in this area and the set of observations presented in the paper is extremely useful for understanding the magnitude of the air pollution problem and the potential sources. However, the language is very vague and the scientific discussion is limited. I find that the improvements are essential before the manuscript can be published in ACP. I list some of the concerns below.

We would like to thank the reviewer for considering the work we have conducted over Lumbini important. We have tried our best to incorporate reviewer's comments and suggestions in the following sections.

I am unsure if what we are most interested in is the comparison of Lumbini measurement with those of other cities. For example, they state that "BC observed at Lumbini was higher by a factor of ~6 and ~4.5 compared to that at Mt. Abu, India and near the base of the Mt. Everest, Nepal, respectively (l.323-326)" but how do we know what to make of these comparisons. What do we learn from these comparisons? To me, it is logical that Lumbini has higher BC

concentrations compared to those remote places. Similarly, I do not find interesting that the ozone concentrations were higher at Lumbini than in the Mt. Everest (l. 332). I would rather be more interested to know how the monthly concentrations change and when the highest and lowest concentration levels were and if there was any difference in the three months or over time among the species.

By comparing with other stations, we intend to show the status of air quality in Lumbini. We have rewritten the section in order to make it more scientific. Moreover, we calculated the monthly concentration change as suggested by the reviewer (see the Figure S5) and added a description in the text (Section 3.2.1). Changes are made in lines 378-400 (lines 394-400 for monthly variation) of the revised manuscript.

Regarding the ozone concentration at Lumbini, we have mentioned that the ozone at Lumbini was (l. 332) found to be lower than at the Mt. Everest region, and have provided the possible reasons for that phenomenon. Please see the Section 3.2.2 in the revised manuscript.
Changes are made in lines 388-393 in the revised manuscript.

Also, examining the period when the model is able to reproduce observations and contrasting that to the times when the model fails would be a good way to make use of both measurements and the model. Such assessment should also provide a good basis for what needs to be improved in the model. I find the argument that the authors put forward on l. 261-262 that the "[D]iscrepancy on model results might have occurred due to various factors inherently uncertain in a weather model" to be hand-waving and not really helpful. With this data set, they should be able to understand the discrepancy between model and the observations a little better.

We thank the reviewer for this suggestion.  As per the advice of other reviewers as well, we now have revised the manuscript to focus more on observation data. The model is used to explain only anthropogenic emission source regions (excluding open burning) using tagged CO tracers. Aerosol modeling studies will be a scope for future paper as model needs improvement (stated in Section 3.2.2).

The manuscript does describe when the model is able to perform well for CO concentration versus times when the predictions are poor (section 3.2.2 and section 3.3.1).

Why is PM1 concentration not discussed in the study (l. 282-283)? If is it discussed elsewhere, please mention it. If there was a problem with the data, then I think this should not be included in this manuscript at all. If there was no problem with the data, I think that can provide an additional insight into the measurements and is worth exploring more than just the average concentrations mentioned in l. 281.

There was no problem with the PM1 data as the instrument provided the concentration of PM10, PM2.5 and PM1 simultaneously. We have included the discussion on PM1 on the revised MS. Reviewer-2 also raised the similar concern. Please see the reply to Reviewer-2 as well on this matter. New additions are shown in lines 273-277, lines 285-296 of the revised manuscript.

I am quite confused about WRF-STEM model simulations. Authors state: "A comparison of model calculated average concentration along with the minimum and maximum concentrations of various pollutants (with observation) is shown in Table 3 (l.340-342)." However, right after this sentence, they write that "[T]he model based concentrations used here are instantaneous values for every third hour of the day (l. 342-343)." Can authors clarify which one that is and if the latter, why did they use the instantaneous values?

Thank you for pointing out this confusion. We mean to say that the model based concentrations were obtained at every third hour unlike the monitoring values. We also would like to make it clear that the modeling values reported in the whole manuscript is every third hour instantaneous value. This is corrected in the revised manuscript.

There are many places where authors state in a very qualitative manner, which obviously is not helpful for the reader to understand the issues being discussed. I list some of the sentences here:

We would, again, like to thank the reviewer for pointing this out. We have tried our best to explain and rephrase the sentences to make it scientific.

1. l.298-299 "BC to CO ratio in Lumbini was found to be different from that observed at other urban and rural sites and those affected by forest fire/biomass burning." What was the ratio observed at Lumbini and other places? What can we infer from this? What is the criterion for "different"?

2. l. 299-302 "a suburban site, Pantnagar, in IGP also observed similar BC to CO ratio." What value is considered "similar" and how is that determined? What do we learn from this?

Here we respond to comment #1 and #2 together. By the sentences as mentioned in l. 298-302, we intend to explain that the ratio of BC to CO observed at Lumbini were different from the ratio obtained over other sites (from literatures) except Pantnagar and Maldives. Reviewer-2 also raised the similar concern. Please see the reply to Reviewer-2 on this matter as well. Changes are shown in lines 324-339 of the revised manuscript.

3. l. 318-321 "PM2.5 concentration in Lumbini have been found to be lower than the megacity like Delhi and north-western IGP regions due to higher level of emissions over those regions." How did they come up with this conclusion? I do not see any comparison of emissions, especially at the sector level. Also, I understood that changing emissions in Lumbini and surrounding regions did not lead to a large concentration difference in the model when they conducted a sensitivity analysis (l. 474-488). Doesn't this conflict with what is argued here?

4. l. 321-323 "BC concentrations observed in Lumbini during pre-monsoon season was lower than the urban Asian cities like Kathmandu and Delhi, slightly higher than in Kanpur but high compared to the remote locations in the region." Are the authors comparing the measurements during the same period between cities? What does "slightly higher" and "high" mean? What is the definition of these? More importantly, what do we learn from this?

Here we respond to comments #3 and #4 together. Unfortunately, we have not conducted any comparison on emission at sector level. We rather conducted comparison on concentrations. We compared the pre-monsoon seasonal average concentrations of BC, PM2.5, CO and $O_3$ obtained at Lumbini with other nearby sites (as mentioned in Table 2). The words like "slightly higher"

and "high" have been removed and the whole paragraph have been rephrased, as already provided in the response to the reviewer's earlier question. Please see lines 382-389 for the changes in the revised manuscript.

5. l. 355-359 "STEM model performance can be significantly improved via better constraining anthropogenic emissions inventory, emissions of open biomass burning and improvements in meteorological output from WRF amongst many other uncertainties inherent in regional chemical transport model." How did they get to this conclusion?

Modeling scope for this current paper has been reduced as per reviewer advice and the associated sentences have been revised throughout the manuscript. However, for the above sentence, pollutant concentration is a function of meteorology (including transport), emissions, and physical and chemical transformation. Many of these processes are parameterized or not accounted in the model. Improvements in any one of these or all of these will lead to improvement in model skills.

6. l. 526-529 "The curve during the prime cooking time is much close to the biomass curve of published data whereas that during non-cooking time is inclined towards the fossil fuel curve." How is "much close" determined, as well as "inclined"?

We have rephrased this sentence in the revised manuscript with a new sentence and provided the comparison figure (Figure 14 in revised manuscript) with published data (to remove the confusion) on biomass and fossil fuel burning as below:

The curve obtained for the prime cooking time is closer towards the published curve on biomass burning whereas that obtained during the non-cooking time is closer towards the published fossil fuel curve.

[Figure]

Figure 14: Comparison of normalized spectral light absorption coefficients obtained during the prime cooking (07:00-08:00 local time) and non cooking time (16:00-17:00 LT) at Lumbini with published data from Kirchstetter et al. (2004).

For the two events when authors found and elevated BC and CO concentrations, what were the PM and O3 levels? Did they find an elevated PM on any other days? Did they find an elevated potassium levels during those days? I think that focusing on the analysis of these two events and clearly explaining the details of the regional contribution assessment presented in the manuscript would definitely strengthen the paper. The regional contribution assessment could be also extended by quantifying the monthly differences and also considering other species. This then could be linked to the chemical composition to assess if the regional contribution has anything to do with the chemical composition difference that they can potentially see in different months.

PM concentrations during two events were 267.33±12.51 (PM10), 107.27±9.20 (PM2.5) and 76.75±7.67 (PM1) µg/m$^3$ during Event-I (i.e., 7-9[th] April, 2013) and 297.60±75.48 (PM10), 117.90±34.85 (PM2.5) and 84.42±25.27 (PM1) µg/m$^3$ during Event-II (i.e., 3-4[th] May, 2013). Similarly O$_3$ concentrations were found to be 53.47(±2.57) and 56.75(±8.35) ppbv during Event-I and Event-II, respectively. The TSP sampling was conducted in coarse resolution (once in 3-6 days) due to which we unfortunately missed the sampling during the events days to evaluate the potassium level during the events.

The chemical composition analysis has been removed from the manuscript as suggested by other reviewers, and it also requires the work beyond the scope of this paper. However, we have added the monthly variation part (in Section 3.3.2) as suggested by the reviewer which reads as:

Regarding the monthly average contribution, the Ganges Valley and Nepal's contribution were almost equal during the month of April (~34% and ~37% respectively) but increased for the Ganges Valley region during the month of May (~44%) in which contribution of Nepal region got reduced (~25%) (Figure S6).

[Figure]

Figure S6: (A) Monthly average model estimated contributions of various source regions to average CO in Lumbini and (B) Time series of region tagged CO tracer during individual months.

Minor comments:

1. rain guage→ rain gauge (l. 239)

   Corrected

2. I'm not sure if the authors really meant the way they wrote the sentence: "But, to our expectation, we could not observe any significant influence of forest fires within the specified grid (l. 419-420)." Did the authors really expect that they would not be able to observe influence? Or is this a typo?

   We would like to thank the reviewer for pointing this out. At first, we were speculating the influence of the forest fires within the specified grid (local forest fire) for the occurrence of events in Lumbini. Later on, when we analyzed the forest fire over the larger region (as shown in Figure 9 and 10 in the ACPD MS), the influence of regional forest fire over our study site was confirmed.

3. Others region→ other regions (l. 469)

   Corrected

References

Kirchstetter, T. W., Novakov, T., and Hobbs, P. V.: Evidence that the spectral dependence of light absorption by aerosols is affected by organic carbon, J. Geophys. Res., 109, D21208, doi:10.1029/2004JD004999, 2004.

**REVIEWER-2**

*General comments*

The paper reports for the first time in Lumbini, Nepal, BC, CO, O3 and PM data from a 3-month experiment in one site. The motivation is to understand air quality in Lumbini, but this objective sounds oversized in regard of the limited duration of the experiment. By the way, no scientific question is set and the methodology presents weak points. Data are new but rather few. No chemical speciation is provided to complete the data of species monitored online. Moreover, those online data could have been further treated: by using ratios (e.g., BC/CO, K/BC, PM1/PM2.5) and the aethalometer model to take full benefit from the BC spectral dependence. The use of modeling is useful to study the synoptic variability of pollutants, but appears highly questionable to simulate the chemical components, given the poor emission data used. A shorter manuscript, attempting to better understand for instance the source effects of the major emission points affecting Lumbini, using data only, not modeling could be considered.

We have tried our best to address the queries and suggestions raised by the reviewer. Scientific question has been provided in the revised manuscript. We feel sorry that we don't have the chemical speciation in this work which is beyond the scope of this paper. The revised manuscript includes substantial discussion using ratios to understand behavior and sources of pollutants. As the reviewer raise concern on the chemical components provided by the model, we have removed compositional analysis portion from the paper in the revised version.

*Specific comments*

Abstract: Objectives and/or a specific question need to be clearly stated.

We have added the objective of our study in the abstract/introduction section.

The main objective of this work was to understand the level of air pollution, diurnal characteristics and the influence of open biomass burning on air quality in Lumbini.

Was any cut-off applied on the BC sampling line, or was it bulk BC?

No specific cut-off was applied on the BC sampling. Hence, the BC concentration reported in the manuscript is total suspended BC (lines 166-167 of the revised manuscript).

It is a pity that PM1 was not considered. The variations of the PM1-to-PM2.5 ratio would possibly provide interesting information about source profiles.

We did not discuss the PM1 concentration in the discussion version of the manuscript. It was not intentional. We have included PM1 now. Based on the suggestions from other reviewers (#1 and 3) as well, we have revised the General overview section (3.2.1) as **General overview, PM ratios and influence of meteorology on pollution concentrations** which now includes the ratios PM1-to-PM2.5, PM2.5-to-PM10, BC-to-PM1 and BC-to-PM2.5 and also the influence of meteorology on concentrations of monitored pollutants. Please see the Section 3.2.1 in the revised manuscript.

Both BC and CO are from incomplete combustion process but the ratio BC/CO is often specific to the different processes. A plot of the variations of BC/CO in time could be more relevant than BC and CO separately.
BC and CO originate from biomass burning as well as from any other fuel and combustion types, as mentioned earlier (line 296). Thus this sentence does not justify the use of BC and CO. Instead, BC/CO could help for source discrimination.

Agree. Regarding the Line 296 (Figure S2), a plot showing the time series of $\Delta BC/\Delta CO$ during the monitoring period has been inserted. In addition, we have also compared the average $\Delta BC/\Delta CO$ ratio observed at Lumbini with other sites (Figure 7) and have included in the main text. The discussions on the new figures have been inserted in the revised version of the manuscript (Section 3.2.1).

Moreover, as suggested by the reviewer, we have replaced the Figure 9 (of ACPD version) with the new figure (Figure 10) provided below where the ratio of $\Delta BC/\Delta CO$ (daily average concentration) has been plotted instead of BC and CO separately.

[Figure]

Figure 10. Time series of daily average ΔBC/ΔCO ratio and absorption Ångstrom exponent (AAE) derived from observations, along with fire counts acquired with the MODIS instrument onboard TERRA satellite for a 200×200 km grid centered at Lumbini. Two rectangular green boxes represent time of two episodes with high peaks in CO and BC concentrations as shown in earlier figures.

Potassium is a biomass burning tracer when the fine fraction is considered. As it can have other sources, it is rather examined as K/BC.

Agree. In the manuscript, we just want to state that the potassium concentration during the pre-monsoon season is higher in the whole year. In order to examine the K/BC ratio, we don't have (at least) a yearlong BC concentration from Lumbini. Recently, our group identified that the $K^+$ in Lumbini is mostly from dust (Wan et al., 2016; manuscript under review for ACPD). So, we have rephrased the sentence in Section 3.3.1 which reads as below:

The chemical composition of TSP filter samples collected at Lumbini also showed higher concentration of Levoglucosan, a biomass burning tracer in Lumbini during the pre-monsoon season as compared to other seasons of the year (Wan et al., 2016). Moreover, Wan et al. (2016) also reported that the highest correlation coefficient between $K^+$ and tracers of dust ($Ca^{2+}$ and $Mg^{2+}$) indicating that dust is the main source of potassium in Lumbini.

Reference:

Wan, X., Kang, S., Li, Q., Rupakheti, D., Zhang, Q., Guo, J., Chen, P., Tripathee, L., Rupakheti, M., Panday, A.K., Wang, W., Kawamura, K., Gao, S., Wu, G. and Cong, Z.: Organic molecular tracers in the atmospheric aerosols from Lumbini, Nepal, in the northern Indo-Gangetic Plain: Influence of biomass burning, Manuscript under review for ACPD, 2016.

Remove PM1

We have discussed the PM1 in the revised MS. So, PM1 has not been removed.

*Technical corrections*

Has been done accordingly in the revised manuscript

"to be the most"

Done

"border"

Changed

"A similar value"

Done latitudes and longitudes, why "s"

"s" removed

"viz." what is the meaning?

"viz." has been replaced with "like"

"gauge"

Changed

"24-hour"

Changed

"the emission inventory shows"

Changed

"15:00" remove "h"

Removed

"is not"

Changed

"these periods"

Changed

**REVIEWER-3**

The authors made a good attempt to conduct the monitoring and modeling studies for the selected air pollutants over the study area. However, the current MS should be further improved before it can be reconsidered for the publication in ACP.

We would like to thank the reviewer for providing useful suggestions to refine our manuscript. Other two reviewers have also pointed out many issues to strengthen our work which we have dealt with in the revised version.

Major comments:
1)      It is not clear what hypothesis the authors want to test in this study hence the content is quite diluted and is difficult to follow the MS.

Through this study we aim to understand and document the level of air pollution in Lumbini, located in the northern edge of the IPG before Himalayan foothills start to rise, during pre-monsoon (significantly polluted season in the Indo-Gangetic Plains),  the diurnal characteristics of various air pollutants, and the influence of open biomass burning on the air quality in Lumbini region.

2)      The linkage between the modeling and monitoring parts appear to be quite weak. How the results of both parts supported each other to reach the study objectives (and what are these?)? If both monitoring and modeling results are to be incorporated then the purpose/research question should be clearly defined from the beginning.

Modeling work is used only to fill data gaps and understand source regions. We have reduced the scope of the modeling work as suggested by reviewers by removing the chemical compositional analysis. Models are evaluated with observations for improvement in emissions inventory and simulating meteorology parameters. Our analysis will aid in improving emissions work. These are highlighted in the manuscript.

The main objective of this work was to understand the level of air pollution, diurnal characteristics and the influence of open biomass burning on air quality in Lumbini.

In my opinion, it would be more interesting if the authors make better attempt to analyze the monitoring data (including also PM1, O3 etc.) in relation to the sources and meteorology, etc. rather than to loosely cover all the activities/results as presented in this version.

We have tried our best to revise the manuscript by discussing more on the observed PM fractions, their ratios, and BC/PM ratio (please see the response to Reviewer-2). Moreover, a new paragraph on influence of meteorology on concentrations of air pollutants (see also Figure S3) has also been inserted. Please see the Section 3.2.1 **General overview, PM ratios and influence of meteorology on pollution concentrations** in the revised manuscript.

3)      The methodology for the modeling part should be described in detail, especially the emission input data. The authors claimed in Line 436 that both modeling and monitoring results showed CO peaks during the biomass burning events but not indicated if and how the emissions from these 2 events were also included in the emission input data.

Section 2.3 does describe the emissions. Anthropogenic emissions are taken from HTAP v2 data while open burning is taken from data from FINN model. Both these emissions are widely used and references for these model/data are cited in the paper. Both emissions are re-gridded to the STEM model domain. Several STEM papers are cited that use this technique. Again due to reviewer's suggestion, the modeling scope of the paper has been reduced to identifying regional sources and filling in data gaps. Thus to go into further model component detail is not warranted.

Minor comments:
1) The description of monitoring instrument (2.2) is lengthy and could be moved to SI.
We have removed the (unwanted) description of monitoring instruments to make the section short and informative.
2) Too many qualitative statements in the MS.

We would like to thank the reviewer for pointing this out. Other reviewer also has pointed the same issue which we have already addressed earlier (please see the responses to the Reviewer-1).

[revised manuscript text omitted]

---

## Referee Report (RR1)

Comments

Overall comment: The authors have improved the MS significantly, especially the monitoring data analysis and interpretation. The major uncertainty however still remains with the modeling part and it was, as pointed out by the authors, mainly due to the emission input data.

Major comments:

- It is not clear how authors extracted/projected the emission provided by EDGAR-HTAP_v2 for the simulation period and how the emissions were segregated temporally (hourly, daily etc.) for the simulation.
- One of the reason of the discrepancy between the modeled and monitored levels is the point-based monitoring as compared to model produced grid average values. However, the model significantly underestimated all species, especially PM. Even CO levels were not reasonably produced as shown in Figure 6, the two events were not reproduced that well (as stated in line 509) as the modeling results appear to be fluctuating continuously during the period.
- Why modeled PM levels are not presented in Figure 6? It is suggested that authors include scatter plots to show the relationship between the monitoring and modeling results for each species in Figure 6. Due to the uncertainty in the model output, all the results and discussion based on the model results may be questionable, i.e. those discussed in Section 3.3.2 (line 524).
- Section 2.3: provide the reasons why this particular model was selected, i.e. if it performs better than other models for the region etc., and how the emission input data was prepared for the modeling period.
- Section 3.1: WRF overestimated temperature and wind speed, underestimated precipitation and RH. Common statistical measures should be used to assess the model performance. For wind direction, because of the circular scale of the measurements (near 0 and near 360 degrees are almost the same) the interpretation of the time series should be made with caution or should be avoided. The comparison should be made for different wind sectors or simply by comparing the modelled windroses with the observed windroses to be presented along in Figure 4.

Suggestion: Since the purpose of using WRF-STEM model was "...to understand pollution source region as well as the contribution of open biomass burning to air quality in Lumbini..." as stated in lines 126-127, it is suggested that authors can use alternative ways of the data analysis to achieve the same aims. For example, analysis of wind field (such as Figure 5) or HYSPLIT trajectories and source locations (hotspots, urban etc.) to show the potential of regional transport of the biomass smoke to the site.

Minor comments:

- Line 180: remove word "dust" because not only dust particles but all the particles
- Line 387: the air pollution levels may not be the right/only criteria to classify an area into semi-urban or rural etc. Please rephrase.
- Line 449: too long a sentence. Please improve the written language.
- Line 617: the content above shows important influence from open burning but the discussion in this section seems to be biased toward residential cooking.

---

## Author Response (AR2)

2017/07/03

Dear Editor,

We would like to express our sincere thanks for serving as the editor for our work.

We would like to inform you that we have revised the manuscript significantly based on the comments by the two reviewers. New interpretation and analysis along with some new figures have been done to answer the questions raised by the reviewers. The "Introduction" section has been shortened based on the suggestions from one of the reviewer. At the same time, the purpose behind incorporating the model in our study has also been defined clearly.

We, at this stage, strongly believe that the quality of the manuscript has improved and look forward to hearing positive result from your side.

Sincerely Yours,

Dipesh Rupakheti and Prof. Shichang Kang, on behalf of all coauthors

**Pre-monsoon air quality over Lumbini, a world heritage site along the Himalayan foothills**

by D. Rupakheti et al.

*Review of Rupakheti et al. (Report #1, Anonymous Referee #4)*

The authors present PM, BC, CO and $O_3$ concentrations measured at Lumbini during April-June 2013 and explained meteorology, pollutant concentrations by conducting WRF-Stem model simulation. They also estimated the regional contributions of CO and aerosol composition to local air quality based only on the model simulation results. This reviewer full agree that the presented observational data set in this study are unique and very useful to understating the level of air pollution in the study area. However, in this revised manuscript, there are several important issues on model simulation and scientific discussion. Therefore, this revised manuscript cannot be accepted in its current form. Before publishing in ACP, several points should be clarified.

We would like to thank the reviewer for his/her constructive comments and suggestions. It seems to us that the reviewer provided suggestions based on the ACPD version (date: 17[th] June, 2016) of the manuscript which makes it difficult to address all of the concerns. For example, based on the earlier reviewers' comments/suggestions, model based aerosol chemical composition has already been removed. However, we have tried our best to accommodate all of the suggestions to the possible extent. Please find the reviewer's comments in black and our replies in blue. The changes in the revised manuscript are colored in red.

*Specific comments and suggestions are below:*

L68, Fig 1: Information given in Figs 1, 2 and 5 are overlapped. This reviewer recommends to merge Fig 1 and Fig 5. That is, plot both monthly mean AOD and winds for separately in April, May and June. Those figures will give more direct insight on aerosol distribution and regional-scale circulation during the intensive measurement period. Fig 2 is not necessary in main body text, so please move it to the supplement.

As suggested by the reviewer, Figure1 and Figure 5 have been merged which is the new Figure 1 in the revised version of the manuscript. Likewise, Figure 2 has been moved to the supplementary information section as Figure S1.

[Figure]

Figure 1. Monthly synoptic wind (at 1000 hPa) for April, May and June 2013, based on NCEP/NCAR reanalysis data where the orientations of arrows refer to wind direction and the length of arrows represents the magnitude of wind (m/s). Red square box in the figure (left) represents the location of Lumbini. Figures on the right column represent monthly aerosol optical depth acquired with the MODIS instrument aboard TERRA satellite. High aerosol loading can be seen over the entire Ingo-Gangetic Plains (IGP). Light gray color used in the figure represents the absence of data.

L68, Fig 1: Which version of MODIS TERRA data have used? Why the authors are not used MODIS Aqua data?

Version 6 (Level 3) of the MODIS TERRA AOD data has been used in this study. We looked at both Aqua and Terra images, which are as follows. They are not significantly different. The difference between the AOD values from two satellites could be due to the true diurnal signal or the retrieval error (Wang et al., 2010).

[Figure]

Figure: Monthly average MODIS Aqua and Terra AOD over the South Asian region during April-June, 2013. The black dot indicates the location of Lumbini.

Reference:

*Wang, L., Wang, Y., Xin, J., Li, Z., & Wang, X. (2010). Assessment and comparison of three years of Terra and Aqua MODIS Aerosol Optical Depth Retrieval (C005) in Chinese terrestrial regions. Atmospheric Research, 97(1), 229-240.*

L54-139: This reviewer recommends to reduce the length of INTRODUCTION section with deleting sentences are not closely related the topic of this paper. For example, the authors emphasize many times in the paper that Lumbini is a UNESCO world heritage. This is interested to the authors, but not to all readers. So please minimize the statements on this.

&

L 99 ~ : As the authors mentioned, sulfuric acid is more critically important in historical heritages. Please more carefully and clearly explain why the air pollutants presented in this study is important should added in the INTRODUCTION.

Agreed. The unwanted length of the Introduction section has been reduced along with the minimal use of information on UNESCO world heritage in this section. The deleted sentences are indicated with the strikethrough. In addition, the significance of the monitored species has been mentioned in the Introduction.

L128: remove "Aerosol optical depth – not discussed on the present study". This is not necessary here.

Agreed and has been removed.

L 165: Table 1 - The sampling period should be more clearly clarified. As shown in Fig. 6, all instruments had not properly operated during the study period.

Done. All instruments ran successfully for the entire duration of the campaign except PM instrument. The following text has been inserted in Section 3.2.1.

*"The gap in the figure (for PM time series) is due to the power interruption to the instrument."*

L169: What's the uncertainty of PM concentration measured with GRIMM EDM164? Especially the quantitative uncertainty of EDM164 for such a high PM concentration level should be discussed, because PM concentration in here is estimated from light scattering measurements.

PM concentrations have been observed highly variable as evident in the standard deviations. For example, variability in $PM_1$, $PM_{2.5}$ and $PM_{10}$ concentration for the month of April is 67.8 %, 60.2 % and 61.8 % respectively from their mean. Similar values were obtained during the month of

May whereas lower values were obtained for the month of June ($PM_1$: 43.6%, $PM_{2.5}$: 45.3 % and $PM_{10}$: 54.1 %). The instrument deployed for PM concentration measurement (GRIMM EDM164) is able to measure the particles mass concentration in the range of 0.1-6000 μg/m³ with the size ranging between 0.25-32 μm with an accuracy of ±5% over the entire measurement range (GRIMM EDM-164 manual; available at : http://wiki.grimm-aerosol.de/images/3/31/GRIMM_EDM_164_datasheet.pdf). Given the high range of the instrument for PM concentration monitoring as compared to that found on our study site, we strongly believe that the data provided by the instrument is highly trustworthy.

Figures 3 & 6 : There is large differences between observations and model simulations. First, more specific explanation and discussion on why the model results were not well agreed with the observations must be addressed. Why the WRF-STEM simulation cannot well simulate the precipitation events and why there is big difference in RH, WD and WS. Why BC is too underestimated compare to the aethalometer data? This should be made for all parameters. This is very important to convincing the results given in Section 3.3 and Section 3.4, as the authors mentioned in L261-262.

The revised version of the paper (submitted on 10th February, 2017) has already addressed this issue. For your information, various sentences on observation and model comparison (on the concentration part) were already removed as suggested by the previous reviewers. In addition, we have applied some general statistics to understand the relationship between observed and modeled species. Please see our response to another reviewer.

Since this is not only modeling based study, it is beyond the scope of the current paper to do sensitivity analysis with different physics scheme or initial and boundary conditions to improve the meteorological prediction. Besides, comparing one station data point with model grid representing 25x25 km is always difficult. We present the comparison results of model to observation to indicate model performance over Lumbini region, not as model validation. We do not have the local emissions inventory and thus we are using the global EDGAR emission in our model. There are plenty of published papers that have used global EDGAR emissions for regional modeling analysis.

Figure 3: WD should NOT be plotted with solid line, because, for example, WD at 355 and 5 degree is almost the same direction. So make a plot with dots.

Agree. The line graph for the WD has been replaced by the dots.

[Figure]

Figure 2. Time series of hourly average observed (red) and model estimated (blue) meteorological parameters at Lumbini, Nepal for the entire measurement period during 1 April to 15 June 2013

L242-243: It is hard to agree to this argument. As shown in Fig. 3, there is large difference between modeled and observed wind speed.

This sentence was already revised during the first revision. We would like to reiterate that the model was able to capture the pattern but not the magnitude; the magnitude by the model is overestimated as compared to the observation which is a common feature of WRF model. Past studies have also proved that the WRF model generally overestimates the WS (Borge et al., 2008; Mohan & Bhati, 2011; Hu et al., 2013; Gunwani & Mohan, 2017).

References

*Borge, R., Alexandrov, V., Del Vas, J. J., Lumbreras, J., & Rodríguez, E. (2008). A comprehensive sensitivity analysis of the WRF model for air quality applications over the Iberian Peninsula. Atmospheric Environment, 42(37), 8560-8574.*

*Mohan, M., & Bhati, S. (2011). Analysis of WRF model performance over subtropical region of Delhi, India. Advances in Meteorology, 2011.*

*Hu, X. M., Klein, P. M., & Xue, M. (2013). Evaluation of the updated YSU planetary boundary layer scheme within WRF for wind resource and air quality assessments. Journal of Geophysical Research: Atmospheres, 118(18).*

*Gunwani, P., & Mohan, M. (2017). Sensitivity of WRF model estimates to various PBL parameterizations in different climatic zones over India. Atmospheric Research, 194, 43-65.*

L258-259: This reviewer also cannot agree to this conclusive sentence. Apparently, the observed RH is two times higher than the modeled one. There is no evidence that RH by model is how well captured the regional variation. Is there a reference data to back this up?

RH values are highly underestimated by the model, however as previously mentioned as in the case of temperature (Section 3.1); the model does not show significant changes in RH during the measurement campaign when the observations stopped working.

L265- Figure 5: How about the winds at 850 hPa or 700 hPa pressure level?

New figures on the winds at 850 hPa have been plotted as follows. Please see the revised text for the discussion.

[Figure]

Figure S3: Wind rose of wind speed and wind direction obtained from the observation (A, B, C) and from the model (D, E, F) for the months of April, May and June 2013 respectively. The right panel shows the synoptic scale wind (850 hPa) during three months of the campaign.

L266-267: Here, what "calm winds" means? This discussion in Figure 6 is conflict the winds discussed in Figures 3 and 4. Please clarify.

"Calm winds" has been replaced by weak winds. The wind direction in Figure 3 (now Figure 2) has been replaced with the dot plot whereas Figure 4 deals with the monthly wind rose plot. However, no weather parameters have been plotted in Figure 6 as the reviewer has indicated. So, we are unable to address this comment.

Figure 6: As commented above, more explanations are needed why there is a large discrepancy between modeled and observed values. Without clarifying this, the results given in the next sections (sections 3.3.2. and 3.4) are not truly reliable.

As previous reviewers have also indicated this, the model output results and associated text have already been revised with unrelated text removed.

L384-385: This reviewer understands the PBL height observation was not available during the measurement period. However, the modeled PBL height has also large uncertainty and not believes it. The authors cited several previous works, but need to add some information on the PBL height, not general seasonal characteristics.

As suggested by the reviewer, we have added following sentences in the PBL description section:

The daily average PBL height obtained from the model is compared with published values (*Wan et al., 2017*) which indicate that the value is captured by our model during initial measurement period and overestimated in the months of mid May onwards. The monthly average diurnal variation also showed that the boundary layer height was maximum during 15:00 local time which coincides with the period of lowest concentration of the pollutants.

[Figure]

Figure 6. Daily time series of PBL height obtained from the model and reported values over Lumbini (obtained from Wan et al., 2017). The lower panel shows the monthly average diurnal variation of the PBL height. The square mark in each box represents the mean PBL height, bottom and top of the box represents 25th and 75th percentile, top and bottom of the whisker represents 90th and 10th percentile respectively.

Reference

*Wan, X., Kang, S., Li, Q., Rupakheti, D., Zhang, Q., Guo, J., . . . Cong, Z. (2017). Organic molecular tracers in the atmospheric aerosols from Lumbini, Nepal, in the northern Indo-Gangetic Plain: Influence of biomass burning. Atmos. Chem. Phys. Discuss., 2017, 1-40. doi:10.5194/acp-2016-1176 (**Accepted for ACP**).*

L407-408: The authors mentioned 'Global Monthly Fire Location Products' were used. However, daily data were used in Figure 9. Please clarify this.

Thanks for pointing out the mistake. The daily data on forest fire were obtained from the FIRMS platform of NASA Earthdata. Correction has been done.

L416: Clearly present how much higher? This is very vague sentence.

 We have revised the sentence as:

*High AAE values (~ 1.6) during these two events are also an indication of presence of BC of biomass burning origin.*

L421: Quality of Figure 10 is very bad. It's hard to read.

We replaced it with high-resolution figure which is now Figure 8. The new figure is given below.

[Figure]

Figure 8. Active fire hotspots in the region acquired with the MODIS instrument on Aqua satellite during (A) Event-I (7-9 April) and (B) Event-II (3-4 May). CO emissions, acquired with AIRS satellite, in the region two days before (3-5 April), during (7-9 April) and two days after (10-12 April) Event-I are shown in panels (C), (E) and (G), respectively while panels (D), (F) and (H) show CO emissions two days before (1-2 May), during (3-4 May) and two days after (5-6 May) the Event-II. Panels (I) and (J) represent the 6-hr interval HYSPLIT back trajectories during Event I and II, respectively. Location of the Lumbini site is indicated by the red star in the panel (I and J). Observed CO versus Model open burning CO illustrating the contribution of forest fires during peak CO loading is shown in panel (K).

Figure 10: How the authors get the modelled biomass CO concentration? Generally the modelled biomass CO concentration is not capture the observed CO concentration. What is the major reason that the author gusse? This reviewer cannot agree to the sentence given in Li435-436.

STEM model can tag the CO emissions originating from biomass burning separately. Comparison of the model to observed CO is done by adding the biomass CO and anthropogenic CO together. This is a standard modeling technique employed by other air quality models as well. Since anthropogenic emissions do not change significantly in a weekly time scale and the temporal variability of the modeled biomass CO matches the temporal variability in the observation, it is inferred that the peak CO events are due to biomass burning. Meteorology could have played a role but we do not see a huge difference in modeled meteorology during the campaign period and observations are not available throughout the campaign.

Figure 10 (I) and (J): Instead of wind roses, regional-scale wind patterns will be more helpful to understand the transport in the interested region.

Thank you for pointing this. From the ground-based observation, we see the local winds coming from the south (for Event-I) whereas the HYSPLIT air mass and synoptic wind both showed that the air mass passed over the fire events in NW IGP. We have replaced the wind rose with the HYSPLIT back trajectories and corrected the text accordingly in section 3.3.1. In addition, the figures for regional-scale wind pattern (during these two events) have been provided in Figure S8 (supplementary materials).

L441-450 and Figure 11: The two ozone peaks were possible contributed by local pollution, induced by NO2, but also by the transport from the fire plume. However, satellite NO2 data shown in Figure 11 is not direct evidence of the effects of fires on high ozone concentration. Clarify this.

It is likely that the local pollution as well as regional pollution (transported from NW IGP region, as indicated by synoptic wind in Fig S8) contributed to the ozone peak. However, we are not able to quantify the individual contributions, even with the model simulation because ozone was not simulated in this experiment. These statements have been added to the text in Section 3.3.1.

L455 and Figure 12: It should be provided how the authors calculated the contributions from different countries? Can you provide PSCF results to back this up? In addition, have the authors estimated other pollutants (i.e., PM2.5 and PM10) for their contribution like CO?

The source regions are identified using CO as a tracer as this is the standard techniques employed by other air quality models (Liu et al., 2003; Price et al., 2003; Pfister et al., 2005; Chen et al., 2009). CO emissions can be tagged according to the country of origin or any given area in the model and subsequently calculate the resulting concentration. The result shown in Figure 10 (current revised version) is done using this methodology. PSCF results are beyond the scope of the current manuscript. We haven't estimated the source regions for other pollutants like PM.

Figure 14: The authors only showed wavelength dependency of normalized light-absorption coefficients for two time periods. First question is why the authors do not show all times? The authors can present with time (x-axis), wavelength (y-axis) and normalized light-absorption coefficient with different color. This figure will be more helpful to understand the difference of light-absorption coefficient in around 380 nm wavelength. Second, the difference at 380 nm in current Figure 14 is statistically significant? And how many data points were used? Last question for Figure 12 is that data during the fire periods discussed section 3.2 were included or excluded here?

The time period selected for Figure 14 (now Figure 11) refer to the periods when the BC concentration was highest and lowest as inferred from the diurnal variation of the BC (Figure 5). Our interest was to study the inclination of the curve during biomass burning dominated period (highest peak in the morning) and fossil fuel dominated period (during afternoon since there is the absence of cooking activities) in Lumbini. We chose those two periods of the day in this study. A new figure has been drawn (shown below) to understand the time series of the normalized light absorption which clearly indicates the highest values of the light absorption at the lowest wavelength. However, to our understanding, this figure possesses difficulties for comparison with reported normalized light absorption values (from literature) which leads us to retain the original figure. But we have included the normalized curve for both of the events which clearly indicated the inclination towards the biomass burning curve. The difference at 380 nm in Figure 11 is statistically insignificant at $p<0.05$. The number of data points used for the cooking and non-cooking periods is 58 for each (excluding the data during two events). Figure 12 does not demonstrate the influence of forest fire, thus we are unable to provide our response to the later part of the query.

[Figure]

Figure: Time series of the normalized light absorption coefficient (normalized at 700nm) observed at Lumbini during the pre-monsoon of 2013.

*Review of Rupakheti et al. (Report #2, Anonymous Referee #3)*

Comments

Overall comment: The authors have improved the MS significantly, especially the monitoring data analysis and interpretation. The major uncertainty however still remains with the modeling part and it was, as pointed out by the authors, mainly due to the emission input data.

We would like to thank the reviewer for his/her comments/suggestions on our work. Please find the reviewer's comments in black and our replies in blue. The changes in the revised manuscript are colored in red.

Major comments:

It is not clear how authors extracted/projected the emission provided by EDGAR-HTAP_v2 for the simulation period and how the emissions were segregated temporally (hourly, daily etc.) for the simulation.

Emissions provided by the EDGAR-HTAP_v2 were re-gridded using four point interpolation technique available in the STEM model emissions preprocessor. The STEM model has been used extensively to study air pollution in Asia. The model has parameterized diurnal emission profile built in the emission preprocessor. Biomass emissions vary on a daily basis as per the burning event detected by FINN model while other anthropogenic emissions are constant over the year without seasonality.

One of the reasons of the discrepancy between the modeled and monitored levels is the point-based monitoring as compared to model produced grid average values. However, the model significantly underestimated all species, especially PM. Even CO levels were not reasonably produced as shown in Figure 6, the two events were not reproduced that well (as stated in line 509) as the modeling results appear to be fluctuating continuously during the period.

Why modeled PM levels are not presented in Figure 6? It is suggested that authors include scatter plots to show the relationship between the monitoring and modeling results for each species in Figure 6. Due to the uncertainty in the model output, all the results and discussion based on the model results may be questionable, i.e. those discussed in Section 3.3.2 (line 524).

The PM composition and quantity discussion are removed in the current version of the manuscript based on the suggestions provided by the previous reviewers. To show improvement in emission inventory and model development, PM comparison statistics between model and observations are shown in Table 3.

Section 2.3: provide the reasons why this particular model was selected, i.e. if it performs better than other models for the region etc., and how the emission input data was prepared for the modeling period.

STEM model has been used extensively to study air pollution in Asia and other parts of the world since its creation in 1987. The model PI, Professor Gregory R. Carmichael has more than 25000 paper citations (Google Scholar Search) mostly based on STEM model studies. We believe that all models have strengths and weaknesses. The authors chose this model because of familiarity with this model.

Emissions provided by the EDGAR-HTAP_v2 were projected using the four point interpolation technique available in the STEM model emissions preprocessor. The model has parameterized diurnal emission profile built in the emission preprocessor. Biomass emissions can vary on a daily basis as per the burning event detected by FINN model while other anthropogenic emissions are constant over the year without seasonality.

Section 3.1: WRF overestimated temperature and wind speed, underestimated precipitation and RH. Common statistical measures should be used to assess the model performance. For wind direction, because of the circular scale of the measurements (near 0 and near 360 degrees are almost the same) the interpretation of the time series should be made with caution or should be avoided. The comparison should be made for different wind sectors or simply by comparing the modelled windroses with the observed windroses to be presented along in Figure 4.

Agree. Based on the suggestion of the reviewer, we calculated correlation, Root Mean Square Error (RMSE) and Mean Absolute Difference (MAD) for the observed and modeled meteorological parameters. The correlation (r) for wind direction, wind speed, temperature and relative humidity were found to be 0.18, 0.22, 0.87 and 0.71 (all values at $P<0.001$) respectively.

Similarly, Root Mean Square Error (RMSE) and Mean Absolute Difference (MAD) were also calculated for the meteorological parameters. RMSE (MAD) values were found as 121.16 (105.67), 3.55 (3.10), 31.66 (27.24), 3.94 (3.28) for wind direction, wind speed, relative humidity and temperature respectively. The values obtained in our study are comparable with those from Delhi during summer as reported in Mohan & Bhati (2011) using the WRF model. Considering the circular scale of the wind direction measurement, we have replaced the line plot previously used for the time series of the wind speed by the dots. This suggestion was provided by another reviewer as well. Now, based on the suggestion by this reviewer, we have plotted the wind rose diagram for the model based values and presented along with the measurement based values as shown in the Figure S3 below. However, please note that comparing wind direction from a point source measurement to a model grid is always difficult. Besides, comparing surface wind direction is more challenging than at higher altitudes where more synoptic winds prevail. In the absence of vertical wind direction observations, we show the surface wind comparison just to indicate model performance, not as model validation. In addition, we also show NCEP/NCAR reanalysis plots in the figure to illustrate the difficulty in comparing wind direction for air pollution transport.

[Figure]

Figure S3: Wind rose of wind speed and wind direction obtained from the observation (A, B, C) and from the model (D, E, F) for the months of April, May and June 2013 respectively. The right panel shows the synoptic scale wind (850 hPa) during three months of the campaign.

[Figure]

Figure 8. Active fire hotspots in the region acquired with the MODIS instrument on Aqua satellite during (A) Event-I (7-9 April) and (B) Event-II (3-4 May). CO emissions, acquired with AIRS satellite, in the region two days before (3-5 April), during (7-9 April) and two days after (10-12 April) Event-I are shown in panels (C), (E) and (G), respectively while panels (D), (F) and (H) show CO emissions two days before (1-2 May), during (3-4 May) and two days after (5-6 May) the Event-II. Panels (I) and (J) represent the 6-hr interval HYSPLIT back trajectories during Event I and II, respectively. Location of the Lumbini site is indicated by the red star in the panel (I and J). Observed CO versus Model open burning CO illustrating the contribution of forest fires during peak CO loading is shown in panel (K).

Reference

*Jaffe, D., Anderson, T., Covert, D., Kotchenruther, R., Trost, B., Danielson, J., ... & Harris, J. (1999). Transport of Asian air pollution to North America. Geophysical Research Letters, 26(6), 711-714.*

Minor comments:

Line 180: remove word "dust" because not only dust particles but all the particles

Done.

Line 387: the air pollution levels may not be the right/only criteria to classify an area into semiurban or rural etc. Please rephrase.

Done. The new sentence now reads as:

*In addition, average BC and CO concentrations in Lumbini were found falling in between concentrations observed at rural sites (up to 6 times higher) and cities in the region (see Table 2), indicating that Lumbini, in a way, can still be considered as semi-urban location.*

Line 449: too long a sentence. Please improve the written language.

 Done. We have rephrased the sentence as:

*Increase in CO concentrations in the evening hours might be due to transport of CO from source regions upwind of Lumbini which along with the local emissions get trapped under reduced Planetary Boundary Layer (PBL) heights.*

Line 617: the content above shows important influence from open burning but the discussion in this section seems to be biased toward residential cooking.

The main aim of this section is to understand the influence of biomass burning on the air quality in Lumbini. Main source of biomass burning in the study area is residential cooking (as shown in Figure 11 and 12 in the current revised version). Due to this fact, we believe that the explanation represents the biomass burning which in a way stands for the residential cooking.

[revised manuscript text omitted]

---

## Author Response (AR3)

**Response to Co-Editor's Comments**

Dear Editor, We would like to express our sincere gratitude for providing the comments and suggestions to improve the quality of our analysis and write up. The response by the authors and changes in the manuscript are presented in 'blue' color.

**Comments**

The MS would benefit much from the improvement of written language to provide sharper discussion of the findings.

Thank You for pointing this issue. In the revised version we have tried our best to remove the redundant sentences as well as improve the writing based on the detailed suggestion provided by the co-editor on the pdf file.

The authors' efforts to compare the results with other studies are appreciable. However, in some places too much comparison without highlighting implications of similarity/difference would dilute the findings.

Thank you for pointing this out. We have mentioned the implications of the past studies we have cited and our analysis to the extent possible. Keeping in mind the suggestion provided by the editor, the literatures which are not directly related to our study have also been removed.

Please elaborate the emission input data for the modeling task, i.e. how the annual emissions (and which year) were interpreted into necessary temporal distributions for the modeling purpose.

Please note that the comparison between observed and modelled in this MS is to test the model performance and it is not for the model VALIDATION. Hence, it is the model "performance" for this case to be discussed and not the model itself (i.e. not the validity of physico-chemical processes incorporated in the model development).

Anthropogenic emission of various pollutants ($CH_4$, CO, $SO_2$, NOx, NMVOC, $NH_3$, $PM_{10}$, $PM_{2.5}$, BC and OC) used in this analysis were taken from the EDGAR-HTAP_v2 for 2010. Annual emissions given in $kg/m^2/sec$ at 0.1x0.1 degree resolution were converted to $molecules/cm^2/sec$ and re-gridded to 25x25 km resolution using four point interpolation techniques available in the STEM emission preprocessor. The emissions were given a diurnal profile using previously used parameterization available in the preprocessor.

We agree that the purpose of the comparison between observed and modeled values is to test the performance rather than model validation. We have replaced the word 'validation' with 'performance'.

The altitude of the model layer, which provided the simulated results for comparison with the observation data, was not given in the MS.

The model data was interpolated to match the observation site's latitude, longitude and altitude for all variables discussed in this paper. This sentence has been inserted in Section 3.1 of the manuscript.

Other minor comments are directly provided in the PDF of the MS. Please note that the comments are inserted in the boxes of the "replace text" function and the MS content needs attention is yellow-marked in the text, references and figure caption.

Thank you for proving detail comments. We have incorporated them.

[revised manuscript text omitted]